# Representational Strengths and Limitations of Transformers

**Clayton Sanford, Daniel Hsu**
Department of Computer Science
Columbia University
New York, NY 10027
{clayton,djhsu}@cs.columbia.edu

**Matus Telgarsky**
Courant Institute
New York University
New York, NY 10012
matus.telgarsky@nyu.edu

## Abstract

Attention layers, as commonly used in transformers, form the backbone of modern deep learning, yet there is no mathematical description of their benefits and deficiencies as compared with other architectures. In this work we establish both positive and negative results on the representation power of attention layers, with a focus on intrinsic complexity parameters such as width, depth, and embedding dimension. On the positive side, we present a *sparse averaging task*, where recurrent networks and feedforward networks all have complexity scaling polynomially in the input size, whereas transformers scale merely *logarithmically* in the input size; furthermore, we use the same construction to show the necessity and role of a large embedding dimension in a transformer. On the negative side, we present a *triple detection* task, where attention layers in turn have complexity scaling linearly in the input size; as this scenario seems rare in practice, we also present natural variants that can be efficiently solved by attention layers. The proof techniques emphasize the value of communication complexity in the analysis of transformers and related models, and the role of sparse averaging as a prototypical attention task, which even finds use in the analysis of triple detection.

## 1 Introduction

In recent years, transformer networks [Vaswani et al., 2017] have been established as a fundamental neural architecture powering state-of-the-art results in many applications, including language modeling [OpenAI, 2023], computer vision [Dosovitskiy et al., 2021], and protein folding [Jumper et al., 2021]. The key building block of transformer models is the *self-attention unit*, a primitive that represents interactions among input elements as inner-products between low-dimensional embeddings of these elements.

The success of transformer models is linked to their ability to scale their training and generalization performance to larger datasets and sequence lengths. Their representational capacity, however, underlies this scaling power, and is tied to the inductive biases of their learning algorithms. Empirically, transformer models trained with gradient-based learning algorithms exhibit biases towards certain algorithmic primitives [Edelman et al., 2022, Liu et al., 2022] and learn representations that may encode domain-specific information in the self-attention units [Clark et al., 2019, Hewitt and Manning, 2019, Rogers et al., 2020, Chen et al., 2022]. These examples indicate that transformer architectures not only provide computational benefits, but also have representational capabilities that are particularly well-matched to practical tasks.

In this paper, we investigate these inductive biases by identifying "natural" computational tasks for which transformers are well-suited, especially compared to other neural network architectures, as well as tasks that highlight the limitations of transformers. The tasks—sparse averaging, pair-matching,

37th Conference on Neural Information Processing Systems (NeurIPS 2023).

and triples-matching—represent primitive operations that aggregate structural information encoded in embeddings. We use these tasks to elucidate the relationship between the embedding dimension $m$ of a self-attention unit and its expressivity, and to showcase the fundamental representational limitations of self-attention layers.

In our model, the primary computational bottleneck faced by a transformer in computing a "sequence-to-sequence"[1] function $f \colon \mathcal{X}^N \to \mathcal{Y}^N$ is the constrained processing of pairs of input elements $\{x_i, x_j\} \in \binom{\mathcal{X}}{2}$; we allow transformers unbounded computational power when processing the individual elements $x_i \in \mathcal{X}$. This is motivated by modern scaling regimes where the context length $N$ has rapidly increased, the self-attention embedding dimension $m$ remains much smaller than $N$, and the parameterization of multi-layer perceptrons (MLPs) that operate on individual elements is much larger than $m$. Indeed, the largest GPT-3 model [Brown et al., 2020] features a context length $N = 2048$, an embedding dimension $m = 128$, and MLPs with a 12288-dimensional parameterization; the context length of GPT-4 is as large as $N = 32000$. As such, we are interested in the capabilities of transformers with $N^{o(1)}$ total "size", as opposed to $N^{\Omega(1)}$. The nature of the bottleneck in our model makes the tools of communication complexity indispensable for formalizing computational limits.

## 1.1 Our contributions

**Sparse averaging separations among atomic self-attention units.** The $q$-*sparse averaging task* $q$SA aims to capture the essential approximation-theoretic properties of self-attention units. In $q$SA, the $i$th input $x_i$ is a pair $(y_i, z_i)$, where $z_i \in \mathbb{R}^{d'}$ is the *data* part of $x_i$, simply a vector in $\mathbb{R}^{d'}$, whereas and $y_i \in \binom{[N]}{q}$ is the *indexing* part, which specifies $q$ locations in the input sequence; the $i$th output element in $q$SA is obtained by averaging the $q$ *data* parts $z_j$ given by $j \in y_i$, meaning

$$q\mathrm{SA}\left((y_1, z_1), \ldots, (y_N, z_N)\right) = \left(\frac{1}{q}\sum_{j \in y_1} z_j, \ldots, \frac{1}{q}\sum_{j \in y_N} z_j\right).$$

(See also Definition 4.) As summarized in the following informal theorem, our analysis of $q$SA in Section 3 and Appendix A illustrates the ability of the self-attention primitive to associate arbitrary subsets of input elements (as opposed to just "local" subsets, as specified by some sequential/topological structure), measures the expressive power accrued by increasing the embedding dimension $m$ of a self-attention unit, and indicates the representational limitations of "traditional" neural architectures on basic computational tasks.

**Informal Theorem 1.** *The task $q$SA for $q \in \mathbb{Z}_+$ satisfies the following properties (see Definition 4 for a formal definition and approximation metric).*

1. *There exists a unit of self-attention $f$ with an $m$-dimensional embedding that approximates $q$SA if and only if $m \gtrsim q$ (Theorems 2 and 4).*

2. *Any fully-connected neural network whose output approximates $q$SA requires its first hidden layer to have width at least $\Omega(Nd)$ (Theorem 10).*

3. *Any recurrent neural network whose iterates approximate $q$SA requires a hidden state of at least $\Omega(N)$ bits (Theorem 11).*

We consider the $q$SA implementation in Item 1 *efficient* since the dimension of the model parameters grows with $\mathrm{poly}(q, d, \log N)$, whereas the latter two are *inefficient* since their parameter (or state) dimension grows as $\mathrm{poly}(N)$. The proofs of the positive results employ embeddings for each index $j$ and each subset $y_i$ that have large inner products if and only if $j \in y_i$. The negative results involve communication complexity reductions and geometric arguments. These arguments naturally introduce a dependence on bits of precision, which we suppress above within the notation "$\gtrsim$"; we note that these bounded-precision results are arguably more relevant to modern networks, which uses as few as 4 or even 2 bits of numerical precision.

**Contrast between pairwise and triple-wise matching with self-attention layers.** We frame standard transformer architectures as being able to efficiently represent functions that are decomposable

---

[1]Note, however, that attention units are permutation equivariant, so the order of elements in the input "sequence" $X \in \mathcal{X}^N$ is irrelevant. In practice, *positional encodings* are used when the sequence order is relevant.

into sparse pairwise interactions between inputs. To do so, we introduce two sequential tasks and prove a collection of constructions and hardness results that characterize the abilities of transformers to solve these tasks.

Given an input sequence $X = (x_1, \ldots, x_N) \in [M]^N$ (for some $M = \text{poly}(N)$), we formalize the problems of *similar pair detection* (Match2) and *similar triple detection* (Match3) as

$$\text{Match2}(X)_{i \in [N]} = \mathbb{1}\left\{\exists j \text{ s.t. } x_i + x_j = 0 \,(\text{mod } M)\right\}, \tag{1}$$

$$\text{Match3}(X)_{i \in [N]} = \mathbb{1}\left\{\exists j_1, j_2 \text{ s.t. } x_i + x_{j_1} + x_{j_2} = 0 \,(\text{mod } M)\right\}. \tag{2}$$

For both tasks, note that the output is an $N$-dimensional vector whose $i$th element is 1 if and only if the sequence $X$ includes a pair or triple *containing $x_i$*. In this sense, the problems differ from 2SUM and 3SUM, which are not sequence-to-sequence tasks.

We believe these two tasks are intrinsically "pairwise" and "triple-wise", respectively; moreover, since we also believe self-attention performs a fundamentally "pairwise" operation, we will use Match2 and Match3 to show a sharp gap in the representation power of self-attention.

**Informal Theorem 2.**

1. *A single unit of standard self-attention with input and output MLPs and an $O(d)$-dimensional embedding can compute* Match2 *(Theorem 6).*

2. *A single layer of standard multi-headed self-attention cannot compute* Match3 *unless its number of heads $H$ or embedding dimension $m$ grows polynomially in $N$ (Theorem 7).*

3. *A standard transformer model* can *efficiently compute a modified version of* Match3 *that makes assumptions about embedding structure or locality (Theorems 8 and 9).*

4. *Under a generalized notion of "third-order tensor self-attention" introduced in Appendix C.3,* Match3 *is efficiently computable with a single unit of third-order attention (Theorem 18).*

While the above result demonstrates the limitations of multi-headed self-attention and illustrates the importance of learning embeddings with contextual clues, we believe that a stronger result exists. Specifically, we conjecture that even multi-layer transformers are unable to efficiently compute Match3 without hints or augmentation.

**Informal Conjecture 1.** *Every multi-layer transformer that computes* Match3 *must have width, depth, embedding dimension, or bit complexity at least $N^{\Omega(1)}$.*

In Appendices C.5 and C.6, we give a heuristic information-theoretic argument to support this conjecture, prove a matching upper-bound, and finally prove analogous results for graph-augmented transformers with respect to the problem of cycle detection in directed and undirected graphs.

## 1.2 Related work

Several computational and learning-theoretic aspects of transformers, distinct from but related to the specific aims of the present paper, have been mathematically studied in previous works.

**Universality and Turing-completeness.** To demonstrate the power of transformers, universal approximation results for transformers [Yun et al., 2020, Wei et al., 2022]—analogous to results for feedforward networks [Hornik et al., 1989]—establish the capability for sufficiently large networks to accurately approximate general classes of functions. Note, however, that the precise minimal dependence of the required size (e.g., number of attention units, depth of the network) as a function of the input size $N$ does not directly follow from such results, and it is complicated by the interleaving of other neural network elements between attention layers. (Approximate) Turing-completeness of transformers demonstrates their power in a different manner, and such results have been established, first assuming infinite precision weights [Pérez et al., 2019] and later also with finite-precision [Wei et al., 2022]. Such results are more closely aligned with our aims, because Turing machines represent a uniform model of computation on inputs of arbitrary size. Wei et al. [2022] showed that Turing machines that run for $T$ steps can be approximated by "encoder-decoder" transformers of depth $\log(T)$ and size polynomial in $\log(T)$ and the number of states of the Turing machine (but the decoder runs for $T$ steps).

**Formal language recognition.** The ubiquity of transformers in natural language understanding has motivated the theoretical study of their ability to recognize formal languages. On the positive side, Bhattamishra et al. [2020] constructed transformers that recognize counter languages, and Yao et al. [2021] showed that transformers of bounded size and depth can recognize Dyck languages that have bounded stack depth. Liu et al. [2022] showed that the computations of finite-state automata on sequences of length $N$ can be performed by transformers of depth $\log(N)$ and size polynomial in the number of states. On the negative side, Hahn [2020] showed limitations of modeling distributions over formal languages (including Dyck) with fixed-size transformers (though this result does not imply quantitative lower bounds on the size of the transformer). Hahn [2020], as well as Hao et al. [2022], also establish the inability of "hard attention" Transformers to recognize various formal languages and circuit classes by leveraging depth reduction techniques from circuit complexity [Furst et al., 1984].

**Learnability.** The sample complexity of learning with low-weight transformers can be obtained using techniques from statistical learning theory and, in turn, establish learnability of certain boolean concept classes (e.g., sparse parity) [Edelman et al., 2022, Bhattamishra et al., 2022] using transformer-based hypothesis classes. Our $q$SA function is inspired by these classes, and we establish concrete size lower bounds for approximation (and hence also learnability) by transformers. We note that our constructions use bounded-size weights, and hence, in principle, the aforementioned sample complexity results can be combined with our results to analyze empirical risk minimization for learning transformers. Prior work of Likhosherstov et al. [2021] also shows how sparse attention patterns can be achieved by self-attention units (via random projection arguments); however, when specialized to $q$SA, their construction is suboptimal in terms of the sparsity level $q$.

**Related models.** Graph neural networks (GNNs), like transformers, process very large inputs (graphs) using neural networks that act only on small collections of the input parts (vertex neighborhoods). Many classes of GNNs are universal approximators for classes of invariant and equivariant functions [Maron et al., 2019, Keriven and Peyré, 2019]. At the same time, they are restricted by the distinguishing power of certain graph isomorphism tests [Xu et al., 2018, Morris et al., 2019, Chen et al., 2019], and lower bounds have been established on the network size to approximate such tests [Aamand et al., 2022]. Loukas [2019] established a connection between GNNs and the LOCAL [Angluin, 1980] and CONGEST [Peleg, 2000] models for distributed computation, and hence directly translates lower bounds for CONGEST—notably cycle detection problems—into size lower bounds for GNNs. Our lower bounds for cycle detection using transformers also leverage a connection to the CONGEST model. However, transformers do not have the same limitations as GNNs, since the computational substrate of a transformer does not depend on the input graph in the way it is with GNNs. Thus, we cannot directly import lower bounds for CONGEST to obtain lower bounds for transformers.

Transformers are also related to other families of invariant and equivariant networks. Our focus on $\mathrm{Match2}$ and $\mathrm{Match3}$ (and related problems) was inspired by the separation results of Zweig and Bruna [2022] between models for processing sets: Deep Sets [Qi et al., 2017, Zaheer et al., 2017], which are "singleton symmetric", and the more expressive Relational Pooling networks [Santoro et al., 2017], which are only "pairwise symmetric".

### 1.3 Conclusion and future work

Our primary contributions are to present a multi-faceted story about transformer approximation: firstly, $q$SA separates transformer models approximation-theoretically from RNNs and MLPs, and moreover the attention embedding dimension both necessary and sufficient for $q$SA scale directly with $q$, meaning $q$SA also functions to characterize representation power amongst different transformers. Secondly, while single units of self-attention can solve the $\mathrm{Match2}$ task, even wide layers of self-attention with high-dimensional embeddings cannot solve $\mathrm{Match3}$, and we believe that deeper models cannot as well. This question of deeper models is stated as a formal conjecture and addressed heuristically in Appendix C.6, using both information- and communication-theoretic proof techniques, both of which we feel are significant steps towards a complete proof.

While our investigation is purely approximation-theoretic, we also include in Appendix D a preliminary empirical study, showing that attention can learn $q$SA with vastly fewer samples than recurrent

networks and MLPs; we feel this further emphasizes the fundamental value of $q$SA, and constitutes an exciting direction for future work.

Beyond the explicit open question in Informal Conjecture 1, we anticipate that future research could connect the separation results proved in this work to formal linguistic theory and empirical work on attention matrix interpretation. This work examines $\mathrm{Match2}$ and $\mathrm{Match3}$ because we believe that the former could represent a key primitive for language processing tasks such as co-referencing, while the latter represents a natural extension of the former that likely is *not* necessary for language modeling. Rather, it may be possible that language modeling performs triple-wise modeling for tasks such as the identification of subject, verb, and object components by relying on pairwise matching constructions and "clues" learned within an embedding, such as those encoded in the toy problems $\mathrm{Match3Bigram}$ and $\mathrm{Match3Local}$. That is, transformers serve as a useful foundational model for language modeling because of their abilities to integrate contextual clues and pairwise communication, and while they are not extensible to "purely triple-wise problems," most practical sequential problems have some efficient decomposition to pairwise structures that can be easily exploited by these architectures. Future work by linguists, theoretical computer scientists, and empirical NLP practitioners could assess how foundational our primitives are and study whether there are any practical triple-wise problems that transformer models fail to solve.

## 2 Preliminaries

Let $\mathbb{B}^d = \{x \in \mathbb{R}^d : \|x\|_2 \leq 1\}$ denote the unit ball in $\mathbb{R}^d$, and let $[n] = \{1, 2, \ldots, n\}$ denote the first $n$ positive integers. The expression $\mathbb{1}\{P\}$ equals $1$ if predicate $P$ is true and $0$ otherwise. The row-wise softmax operator applied to matrix $A \in \mathbb{R}^{N \times M}$ returns

$$\mathrm{softmax}(A)_{i,j} = \frac{\exp(A_{i,j})}{\sum_{j'=1}^{M} \exp(A_{i,j'})}.$$

### 2.1 Attention units and transformer architectures

We first introduce the concept of self-attention, which is used as the building block of all transformer architectures included in this paper.

**Definition 1.** For input dimension $d$, output dimension $d'$, embedding dimension $m$, precision $p$, and matrices $Q, K \in \mathbb{R}^{d \times m}$ and $V \in \mathbb{R}^{d \times d'}$ (encoded using $p$-bit fixed-point numbers), a *self-attention unit* is a function $f_{Q,K,V} : \mathbb{R}^{N \times d} \to \mathbb{R}^{N \times d'}$ with

$$f_{Q,K,V}(X) = \mathrm{softmax}(XQK^\mathsf{T}X^\mathsf{T})XV.$$

Let $\mathcal{A}_{d,m,d',p} = \{f_{Q,K,V} : Q, K, V\}$ denote all such self-attention units.

Self-attention units can be computed in parallel to create multi-headed attention.

**Definition 2.** For head-count $H$ and self-attention units $f_1, \ldots, f_H \in \mathcal{A}_{d,m,d',p}$, a *multi-headed attention layer* is a function $L_{f_1,\ldots,f_H} : \mathbb{R}^{N \times d} \to \mathbb{R}^{N \times m}$ with $L_{f_1,\ldots,f_H}(X) = \sum_{h=1}^{H} f_h(X)$. Let $\mathcal{A}_{d,m,d',p}^{H}$ contain all such $L_{f_1,\ldots,f_H}$.

Transformer models are composed of two components: multi-headed attention layers (as above) and element-wise multi-layer perceptrons. Due to universal approximation results, we model multi-layer perceptrons as arbitrary functions mapping fixed-precision vectors to themselves.

**Definition 3.** A *multi-layer perceptron (MLP) layer* is represented by some $\phi : \mathbb{R}^d \to \mathbb{R}^{d'}$, whose real-valued inputs and outputs can be represented using $p$-bit fixed-precision numbers. We apply $\phi$ to each element (i.e., row) of an input $X \in \mathbb{R}^{N \times d}$, abusing notation to let $\phi(X) = (\phi(x_1), \ldots, \phi(x_N)) \in \mathbb{R}^{N \times d'}$. Let $\Phi_{d,d',p}$ denote all such MLPs.

We concatenate the notation of each class of functions to denote function composition. For example, for output dimension $d'$, we use $\mathcal{A}'_{d,m,d',p} := \mathcal{A}_{m,m,d',p}\Phi_{d,m,p}$ and $\mathcal{A}^{H\prime}_{d,m,d',p} := \mathcal{A}^{H}_{m,m,d',p}\Phi_{d,m,p}$ to represent single-headed and multi-headed attention units with an input MLP respectively. (The capabilities and limitations of these models are studied in Section 3.) For depth $D$, we let

$$\mathcal{T}_{d,m,d',p}^{D,H} = \Phi_{m,d',p}(\mathcal{A}_{m,m,m,p}^{H\prime})^{D-1}\mathcal{A}_{d,m,m,p}^{H\prime}$$

represent a full transformer model comprising $D$ layers of $H$-headed self-attention with interspersed MLPs.

While two key features of transformer architectures—the residual connection and the positional embedding—are conspicuously missing from this formalism, the two can be implemented easily under the framework. We can include a positional embedding by encoding the index as a coordinate of the input, i.e. $x_{i,1} = i$. Then, the subsequent MLP transformation $\phi(X)$ can incorporate $i$ suitably into the embedding. A residual connection can be included additively as input to a multi-layer perceptron layer (as is standard) by implementing an "approximate identity" attention head $f$ with $Q, K$ and $V = I_m$ set to ensure that $f(X) \approx X$.[2]

We periodically consider transformers implemented with real-valued arithmetic with infinite bit complexity; in those cases, we omit the bit complexity $p$ from the notation.

Finally, we assume for the proof of Theorem 3 that the model is permitted to append a single <END> token at the end of a sequence. That is, we say that a model $f \in \mathcal{T}_{d,m,d',p}^{D,H}$ represents a target $h : \mathbb{R}^{N \times d} \to \mathbb{R}^{N \times d'}$ if $f(X')_{1:N} = g(X)$ when $X' = (x_1, \ldots, x_N, x')$ for constant-valued $x' \in \mathbb{R}^d$.

## 3 Sparse averaging with attention units

We present the sparse averaging task to highlight the ability of transformer architectures to simulate a wide range of meaningful interactions between input elements. This task demonstrates how the embedding dimension of a self-attention unit modulates the expressive capabilities of the architecture, while showcasing the inabilities of fully-connected and recurrent neural networks to capture similar interactions (see Appendix A).

**Definition 4.** For sparsity $q$, problem dimension $d'$, and input dimension $d = d' + q + 1$, consider an input $X = (x_1, \ldots, x_N) \in \mathbb{R}^{N \times d}$ with $x_i = (z_i; y_i; i)$ for $z_i \in \mathbb{B}^{d'}$ and $y_i \in \binom{[N]}{q}$.[3] Let the *q-sparse average* be

$$q\mathrm{SA}(X) = \left( \frac{1}{q} \sum_{j=1}^{q} z_{y_{i,j}} \right)_{i \in [N]}.$$

For accuracy $\epsilon > 0$, a function $f : \mathbb{R}^{N \times d} \to \mathbb{R}^{N \times d'}$ *$\epsilon$-approximates* $q\mathrm{SA}$ if for all $X$,

$$\max_{i \in [N]} \|f(X)_i - q\mathrm{SA}(X)_i\|_2 \leq \epsilon.$$

Figure 1a visualizes the sparse averaging task as a bipartite graph between subsets $y_i$ and elements $z_i$ with corresponding averages. Theorems 2 and 4 jointly show that the minimum embedding dimension $m$ of single self-attention units $\mathcal{A}'_{d,m,d',p}$ that $O(\frac{1}{q})$-approximate $q\mathrm{SA}$ scales linearly with $q$. We believe that the sparse averaging problem is thus a canonical problem establishing the representational capabilities and inductive biases of self-attention units.

### 3.1 Self-attention can approximate $q\mathrm{SA}$ when $m \gtrsim q$

Our principle positive result shows that the sparse averaging task $q\mathrm{SA}$ can be approximately solved using fixed-precision arithmetic self-attention units with embedding dimension $m$ growing with $q \log N$.

**Theorem 2** (Fixed-precision). *For any $N$, any $m \geq \Omega(d' + q \log N)$, any $\epsilon \in (0,1)$, and $p = \Omega(\log(\frac{q}{\epsilon} \log N))$, there exists some $f \in \mathcal{A}'_{d,m,d',p}$ that $\epsilon$-approximates $q\mathrm{SA}$.*

While the full proof appears in Appendix B.1, we briefly sketch the argument here. Because the output of a self-attention unit is a convex combination of rows of the value matrix $\phi(X)V \in \mathbb{R}^{N \times d'}$, a natural way to approximate $q\mathrm{SA}$ with a unit of self-attention is to let each value be the corresponding

---

[2]A simple construction involves letting $XQ = XK$ with iid Gaussian columns fixed for every index $i$. Then, the diagonals of $XQK^\top X^\top$ are far larger than all other entries and its softmax is approximately $I_N$.

[3]We may encode a $q$ element subset of $[N]$ as a vector in $[N]^q$ constrained to have distinct components.

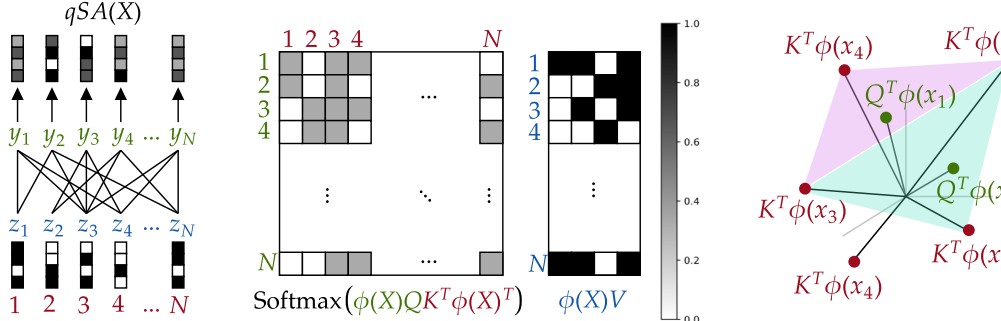

(a) Bipartite graph relating $y_i$ and $z_i$ in $q\mathrm{SA}(X)$.

(b) Attention and value matrices used for the self-attention construction of $q\mathrm{SA}(X)$ in Theorem 2.

(c) Key and query embeddings that produce the self-attention matrix in (b).

Figure 1: A visualization of the $q\mathrm{SA}$ function outputs given a sequence of inputs $(z_i; y_i; i)_{i \in [N]}$ as a bipartite graph between subsets $y_i$ and vectors $z_i$ (a), and of the attention matrix (b) and underlying embeddings (c) that produce the self-attention construction in Theorem 2.

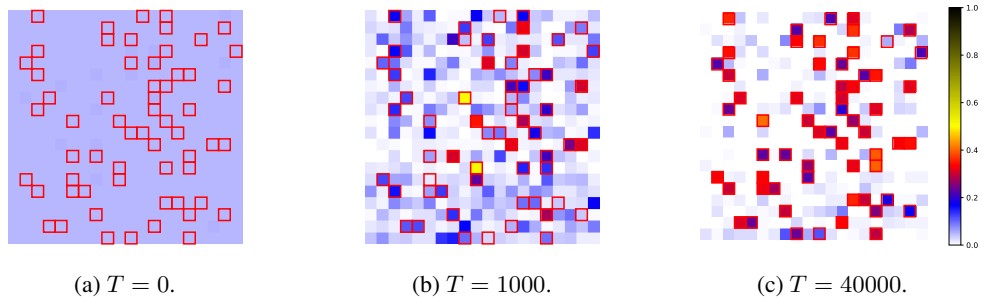

(a) $T = 0$.  (b) $T = 1000$.  (c) $T = 40000$.

Figure 2: Attention matrix $\mathrm{softmax}(\phi(X)QK^\intercal\phi(X)^\intercal) \in \mathbb{R}^{20 \times 20}$ for a fixed example after $T$ epochs of training a self-attention unit to solve $q\mathrm{SA}$ for $q = 3$. Each row $i$ corresponds to subset $y_i$, and each cell $j \in y_i$ is outlined in red. See Appendix D for experimental details.

vector in the average (i.e. $V^\intercal\phi(x_i) = z_i$) and choose the key and query functions in order to ensure that the attention matrix satisfies

$$\mathrm{softmax}(\phi(X)QK^\intercal\phi(X)^\intercal)_{i,j} \approx \begin{cases} \frac{1}{q} & \text{if } j \in y_i, \\ 0 & \text{otherwise.} \end{cases}$$

To do so, let each key $K^\intercal\phi(x_i)$ represent a fixed vertex on a convex polytope, which depends only on index $i$ and is constructed from random binary vectors. We select each query $Q^\intercal\phi(x_i)$ to ensure that $\phi(x_i)^\intercal QK^\intercal\phi(x_j)$ is a fixed large value if $j \in y_i$ and a slightly smaller value otherwise. We obtain the precise query, key, and value embeddings by employing tools from dual certificate analysis from the theory of compressed sensing.

We visualize this construction in Figure 1b and 1c for $q = 3$ and $d' = 4$, which presents the associated attention and value matrices necessary for the construction, and plots a polytope of keys (red dots) with each face corresponding to each subset $y_i$ (green dots). The construction is empirically relevant; Figure 2 shows that a unit of self-attention trained on data generated by the $q\mathrm{SA}$ task recovers a similar attention matrix to the one stipulated in our construction and visualized in Figure 1b.

The logarithmic dependence of the embedding dimension $m$ on the sequence length $N$ can be eliminated by considering self-attention units with real-valued arithmetic with infinite bit complexity.

**Theorem 3** (Infinite-precision). *For fixed $N$, $m \geq \Omega(d'+q)$ and $\epsilon > 0$, there exists some $f \in \mathcal{A}'_{d,m,d'}$ that $\epsilon$-approximates $q\mathrm{SA}$.*

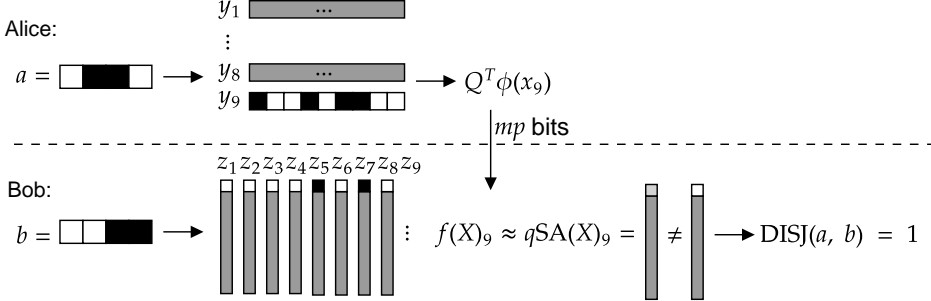

Figure 3: The $mp$-bit communication protocol used to reduce the hardness of computing $q$SA with a single unit of self-attention to the hardness of solving the DISJ communication problem for the proof of Theorem 4 for $q = 4$.

The proof of Theorem 3 employs a similar polytope-based construction in Appendix B.2, relying on a cyclic polytope rather than one drawn from discrete boolean vectors. Theorem 16 proves the near-optimality of *that* bound by employing a geometric argument to show that a variant of $q$SA can only be approximated by a restricted family of self-attention units with a sufficiently high-dimensional embedding.

## 3.2 Self-attention cannot approximate $q$SA when $m \lesssim q$

We show that the construction used to prove Theorem 2 is nearly optimal.

**Theorem 4.** *For any sufficiently large $q$, any $N \geq 2q + 1$, and any $d' \geq 1$, there exists a universal constant $c$ such that if $mp \leq cq$, then no $f \in \mathcal{T}_{d,m,d',p}^{1,1}$ exists that $\frac{1}{2q}$-approximates $q$SA.*

(By choosing $p = O(\log(q \log N))$, Theorem 2 is shown to be optimal up to logarithmic factors of $q$ and doubly-logarithmic factors of $N$.)

The proof of Theorem 4 employs a standard communication complexity argument based on a reduction from the following *set disjointness* problem in the two-party communication model, in which each party possesses a subset of an $n$ element domain (encoded as $n$-bit strings), and they wish to jointly determine whether their subsets are disjoint. We note that communication complexity is commonly-used technique for proving lower bounds on the representational power of circuits and feedforward neural networks [see, e.g., Karchmer and Wigderson, 1988, Ben-David et al., 2002, Martens et al., 2013, Vardi et al., 2021].

**Fact 5** (Set disjointness communication lower bound [Yao, 1979]). *Suppose Alice and Bob are given inputs $a, b \in \{0, 1\}^n$, respectively, with the goal of jointly computing $\mathrm{DISJ}(a, b) = \max_i a_i b_i$ by alternately sending a single bit message to the other party over a sequence of communication rounds. Any deterministic protocol for computing $\mathrm{DISJ}(a, b)$ requires at least $n$ rounds of communication.*

Our proof designs a communication protocol that Alice and Bob use to jointly compute $\mathrm{DISJ}(a, b)$ when $n = q$ in $O(mp)$ rounds of communication, under the assumption that such an $f$ exists that closely approximates $q$SA.

- Alice encodes her input $a$ in a single subset by letting $y_{2q+1} = \{2i + a_i - 1 : i \in [q]\}$.

- Bob uses his input $b$ to assign $z_{2i-1}$ to $2b_i - 1$ and $z_{2i} = -1$ for all $i \in [q]$.

- All other input components are set to constant values known by both parties.

Alice sends her $mp$-bit query embedding $Q^\top \phi(x_{2q+1})$ bit-by-bit to Bob, who approximately computes $q$SA by determining the outcome of $f$. The crux of the reduction shows that $q\mathrm{SA}(X)_{2q+1} = -1$ if and only if $a_i b_i = 0$ for all $i \in [q]$, which allows Bob to determine $\mathrm{DISJ}(a, b)$.

We visualize the protocol in Figure 3 and give the proof in Appendix B.3. The proofs of Theorems 7, 11, 21, and 23 employ similar communication complexity reductions to DISJ.

# 4 Standard transformer models can only efficiently represent intrinsically pairwise functions

In this section, we argue that the standard transformer architecture is unable to efficiently represent functions that do not decompose into a small number of pairwise-symmetric functions. We do this by contrasting the (in)approximability of intrinsically pairwise and triple-wise functions, respectively Match2 and Match3 (defined in (1) and (2)), and their variants.

## 4.1 Efficient computation of Match2 with standard self-attention

We first show that Match2 can be efficiently approximated by a single standard (pairwise) self-attention unit.

**Theorem 6.** *For any input size $N$, input range $M = N^{O(1)}$, and fixed-precision bit complexity $p = O(\log M)$, there exists a transformer architecture $f \in \mathcal{T}_{N,m,1,p}^{1,1}$ with a single self-attention unit with embedding dimension $m = 3$ such that for all $X \in [M]^N$, $f(X) = \text{Match2}(X)$.*

The proof, given in Appendix C.1 uses both a "blank token" and a trigonometric positional embedding, which ensures that

$$\phi(x_i)^\intercal Q K^\intercal \phi(x_j) = c \sum_{k=1}^{d} \cos\left(\frac{2\pi(x_{i,k} + x_{j,k})}{M}\right)$$

for some sufficiently large constant $c$. This embedding ensures that a cell of the attention matrix $\text{softmax}(\phi(X) Q K^\intercal \phi(X)^\intercal)_{i,j}$ is extremely close to zero, unless $x_i = -x_j \pmod{M}$.

## 4.2 Hardness of computing Match3 with a multi-headed self-attention layer

Although Match2 can be efficiently represented using a single unit of standard self-attention, representing Match3 using an entire layer of multi-headed attention units is impossible unless either the number of heads $H$, the embedding dimension $m$, or the precision $p$ grows as $N^{\Omega(1)}$.

**Theorem 7.** *There is universal constant $c > 0$ such that for sufficiently large $N$, and any $M \geq N + 1$, if $mpH \leq cN/\log\log N$, then there is no $f \in \mathcal{T}_{1,m,1,p}^{1,H}$ satisfying $f(X) = \text{Match3}(X)$ for all $X \in [M]^N$.*

We give the proof in Appendix C.2. Like that of Theorem 4, the proof relies on a reduction from set disjointness in two-party communication. The proof of the lower bound applies a domain-restricted variant of Match3, which actually makes the problem substantially simpler to solve. In Remark 1, we show how this variant of Match3 introduces a *depth separation* between the representational powers of single-layer and two-layer transformer models.

As mentioned in the introduction, we also conjecture that multiple layers of multi-headed attention are subject to the same impossibility (Conjecture 19). The impossibility is specific to standard (pairwise) attention; in Appendix C.4, we show that Match3 *can* be efficiently computed with a single unit of *third-order* self-attention.

## 4.3 More efficient constructions for simplified Match3 computations

While the previous sections suggests that no efficient construction exists to compute Match3 with standard transformer models, practical examples of triple detection abound. For example, a transformer-based language model will likely succeed in linking a subject/verb/object triple because all three tokens likely inhabit the same local region and because the model could agglomerate the triple by first identifying a pair and then adding the third. Here, we introduce two variants on the Match3 problem that have additional structure to serve as hints. The first variant specifies triple sums comprising the input element and a neighboring pair elsewhere in the sequence: for each $i \in [N]$,

$$\text{Match3Bigram}(X)_i = \mathbb{1}\left\{\exists j \text{ s.t. } x_i + x_j + x_{j+1} = 0 \,(\text{mod } M)\right\}.$$

The second focuses on localized sums, where are all components of a triple must be within a fixed range of constant width $K \ll N$: for each $i \in [N]$,

$$\text{Match3Local}(X)_i = \mathbb{1}\left\{\exists j_1, j_2 \text{ s.t. } x_i + x_{j_1} + x_{j_2} = 0 \,(\text{mod } M), |i - j_1|, |i - j_2| \leq K\right\}.$$

We show that the two can be efficiently represented using compact standard transformer models.

**Theorem 8.** *For any $N$, $M = N^{O(1)}$, and $p = O(\log M)$, there exists a transformer architecture $f \in \mathcal{T}_{1,m,1,p}^{D,1}$ with embedding dimension $m = 3$ and depth $D = 2$ such that for all $X \in [M]^{N \times d}$, $f(X) = \mathrm{Match3Bigram}(X)$.*

Informally, the first layer of the construction uses a sinusoidal positional encoding to compute each bigram sum $x_j + x_{j+1}$ in the $j$th element of the sequence. The second layer applies the Match2 construction provided by Theorem 6 to determine whether there exists a $j$ for each $i$ such that $x_i + x_j + x_{j+1} = 0 \pmod{M}$.

**Theorem 9.** *For any $d$, $N$, $M = N^{O(1)}$, $p = O(\log M)$, and $K \leq N$, there exists a transformer architecture $f \in \mathcal{T}_{1,m,1,p}^{1,1}$ with embedding dimension $m = O(K \log N)$ and bit-complexity $p = O(\log(K \log N))$ such that for all $X \in [M]^{N \times d}$, $f(X) = \mathrm{Match3Local}(X)$.*

*Proof.* We implement the localized construction by using Theorem 2 to construct a specific sparse simultaneous average of the inputs with $q := 2K + 1$ and $d' := 2K + 1$. To do so, we use the input MLP to convert $x_i$ to the embedding $(z_i; y_i; i)$, for zero-padded input

$$z_i = x_i e_{\bar{i}} \in \mathbb{R}^{2K+1}$$

for $\bar{i} = i \pmod{2K + 1}$ and subset

$$y_i = \{i - K, i - K + 1, \ldots, i + K\} \in \binom{[N]}{2K + 1}.$$

This construction ensures that the $i$th element of self-attention output computes (a rotation of) $(x_{i-K}, x_{i-K+1}, \ldots, x_{i+K})$. An output MLP can then verify whether any matching triples involving $x_i$ exist among those vectors. $\square$

## Acknowledgments and Disclosure of Funding

We are grateful for many discussions with and feedback from Navid Ardeshir, Peter Bartlett, Alberto Bietti, Yuval Efron, Christos Papadimitriou, Shivam Nadimpalli, Rocco Servedio, Yusu Wang, and Cyril Zhang. This work was supported in part by NSF grants CCF-1740833 and IIS-1563785, a JP Morgan Faculty Award, and an NSF Graduate Research Fellowship.

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

# A Fully-connected neural networks and recurrent neural networks cannot efficiently approximate $q$SA

## A.1 Only wide fully-connected neural networks can approximate $q$SA

In this section, we show that any fully-connected neural network that approximates $q\text{SA} : \mathbb{R}^{Nd} \to \mathbb{R}^{Nd'}$ must have width $m = \Omega(N)$.[4] We consider networks of the form $f(x) = g(Wx)$ for some weight matrix $W \in \mathbb{R}^{m \times Nd}$ (the first layer weights) and arbitrary function $g : \mathbb{R}^m \to \mathbb{R}^{Nd'}$ (computed by subsequent layers of a neural network).

**Theorem 10.** *Suppose $q \leq \frac{N}{2}$. Any fully-connected neural network $f$ defined as above that $\frac{1}{2q}$-approximates $q$SA satisfies $m \geq \text{rank}(W) \geq \frac{Nd'}{2}$.*

*Proof.* For simplicity, we arrange the input as

$$x = (1; \ldots; N; y_1; \ldots; y_N; z_1; \ldots; z_N)$$

and $W = [\tilde{W}; V_1; \ldots; V_N]$ with $z_1, \ldots, z_N \in \mathbb{B}^{d'}$, $\tilde{W} \in \mathbb{R}^{m \times N(d-d')}$, and $V_1, \ldots, V_N \in \mathbb{R}^{m \times d'}$. If $\text{rank}(W) \leq \frac{Nd'}{2} - 1$, then so too is $\text{rank}([V_q; \ldots; V_N]) \leq \frac{Nd'}{2} - 1$, and $[V_q; \ldots; V_N]$ has a nontrivial null space containing a nonzero vector $u = (u_q; \ldots; u_N) \in \mathbb{R}^{(N-q)d'}$. Let

$$\xi = \frac{1}{\max_{j \in \{q, \ldots, N\}} \|u_j\|_2} (u_q; \ldots; u_N),$$

$z = (\vec{0}; \ldots; \vec{0}; \xi_q; \ldots; \xi_N)$, and $z' = (\vec{0}; \ldots; \vec{0}; -\xi_q; \ldots; -\xi_N)$. Then,

1. $z_j, z'_j \in \mathbb{B}^{d'}$ for all $j \in [N]$;

2. $V_j z_j = V_j z'_j = 0$ for all $j \in [N]$; and

3. $\|z_{j^*} - z'_{j^*}\|_2 = 2$ for some $j^* \in \{q, \ldots, N\}$.

Therefore, for any $y_1, \ldots, y_N \in \binom{[N]}{q}$, respective $x = (1; \ldots; N; y_1; \ldots; y_N; z_1; \ldots; z_N)$ and $x' = (1; \ldots; N; y_1; \ldots; y_N; z'_1; \ldots; z'_N)$ satisfy $f(x) = f(x')$. Consider $y$ with $y_j = (1, \ldots, q-1, j)$ for each $j \in \{q, \ldots, N\}$. Then,

$$q\text{SA}(x)_j = \frac{1}{q} \xi_j \text{ and } q\text{SA}(x')_j = -\frac{1}{q} \xi_j.$$

Hence, $\|q\text{SA}(x)_{j^*} - q\text{SA}(x')_{j^*}\|_2 \geq \frac{2}{q}$. Because $f(x) = f(x')$,

$$\max \left( \|f(x) - q\text{SA}(x)_{j^*}\|_2, \|f(x') - q\text{SA}(x')_{j^*}\|_2 \right) \geq \frac{1}{q},$$

so $f$ can approximate $q$SA to accuracy no better than $\frac{1}{q}$. $\qquad\qquad\square$

## A.2 Only high-memory recurrent neural networks can approximate $q$SA

In this section, we show that any memory-bounded algorithm that approximates $q\text{SA} : \mathbb{R}^{N \times d} \to \mathbb{R}^{N \times d'}$ must use a large "hidden state" (memory) as it processes the input elements. This lower bound applies to various recurrent neural network (RNN) architectures.

A memory-bounded algorithm with an $m$-bit memory processes input $X \in \mathbb{R}^{N \times d}$ sequentially as follows. There is an initial memory state $h_0 \in \{0, 1\}^m$. For $i = 1, 2, \ldots, N$, the algorithm computes the $i$-th output $f(X)_i \in \mathbb{R}^{d'}$ and the updated memory state $h_i$ as a function of the input $x_i \in \mathbb{R}^d$ and previous memory state $h_{i-1}$:

$$(f(X)_i, h_i) = g_i(x_i, h_{i-1}),$$

---

[4]We regard inputs as $Nd$-dimensional vectors rather than $N \times d$ matrices.

where $g_i \colon \mathbb{R}^d \times \{0,1\}^m \to \mathbb{R}^{d'} \times \{0,1\}^m$ is permitted to be an arbitrary function, and $f \colon \mathbb{R}^{N \times d} \to \mathbb{R}^{N \times d'}$ is the function computed by the algorithm.

Our lower bound applies to algorithms that only need to solve the subclass of "causal" instances of $q$SA in which the input $X = ((z_i, y_i, i))_{i \in [N]} \in \mathbb{R}^{N \times d}$ is promised to satisfy $y_i = \emptyset$ for all $i \le N/2 + 1$, and $y_i \subseteq \{1, \ldots, N/2 + 1\}$ for all $i > N/2 + 1$.

**Theorem 11.** *For any $\varepsilon \in (0, 1)$, any memory-bounded algorithm that $\varepsilon$-approximates $q$SA (for $q = 1$ and $d' = 1$) on the subclass of "causal" instances must have memory $m \ge (N - 1)/2$.*

*Proof.* Consider an $m$-bit memory-bounded algorithm computing a function $f \colon \mathbb{R}^{N \times d} \to \mathbb{R}^N$ that $\varepsilon$-approximates $q$SA (for $q = 1$ and $d' = 1$). We construct, from this algorithm, a communication protocol for DISJ (with $N = 2n + 1$) that uses $m$ bits of communication.

Let $a, b \in \{0, 1\}^n$ be the input for DISJ provided to Alice and Bob, respectively. The protocol is as follows.

1. Alice constructs inputs $x_i = (z_i, \emptyset, i)$ for $i = 1, \ldots, n+1$, where for each $i = 1, \ldots, n$,

$$z_i = \begin{cases} +1 & \text{if } a_i = 0, \\ -1 & \text{if } a_i = 1, \end{cases}$$

   and

$$z_{n+1} = +1.$$

   Bob constructs inputs $x_{n+1+i} = (0, y_{n+1+i}, n+1+i)$ for $i = 1, \ldots, n$, where

$$y_{n+1+i} = \begin{cases} \{n+1\} & \text{if } b_i = 0, \\ \{i\} & \text{if } b_i = 1. \end{cases}$$

   Observe that, for this input $X = (x_1, \ldots, x_{2n+1})$, we have

$$q\mathrm{SA}(X)_{n+1+i} = \begin{cases} +1 & \text{if } a_i b_i = 0, \\ -1 & \text{if } a_i b_i = 1. \end{cases}$$

2. Alice simulates the memory-bounded algorithm on the first $n+1$ inputs $x_1, \ldots, x_{n+1}$, and sends Bob the $m$-bit memory state $h_{n+1}$. This requires $m$ bits of communication.

3. Starting with $h_{n+1}$, Bob continues the simulation of the memory-bounded algorithm on these $n$ additional inputs $x_{n+2}, \ldots, x_{2n+1}$.

4. If any output $f(X)_{n+1+i}$ for $i = 1, \ldots, n$ satisfies

$$f(X)_{n+1+i} < 0,$$

   then Bob outputs 1 (not disjoint); otherwise Bob outputs 0 (disjoint).

The approximation guarantee of $f$ implies that $\mathrm{sign}(f(X)_{n+1+i}) = q\mathrm{SA}(X)_{n+1+i}$ for all $i = 1, \ldots, n$, so Bob outputs 1 if and only if $a$ and $b$ are not disjoint. Because this protocol for DISJ uses $m$ bits of communication, by Fact 5, it must be that $m \ge n = (N - 1)/2$. $\square$

We note that the proof of Theorem 11 can be simplified by reducing from the INDEX problem, which has a 1-way communication lower bound of $n$ bits. This suffices for "single pass" algorothms, such as standard RNNs. However, the advantage of the above argument (and reducing from DISJ) is that it easily extends to algorithms that make multiple passes over the input. Such algorithms are able to capture bidirectional recurrent neural net and related models. A straightforward modification of the protocol in the proof of Theorem 11 shows that $\Omega(N)$ memory is required for any algorithm that makes $O(1)$ passes over the input (and computes the outputs in a final pass).

# B  Supplementary results for Section 3

## B.1  Proof of Theorem 2

**Theorem 2** (Fixed-precision). *For any $N$, any $m \geq \Omega(d' + q \log N)$, any $\epsilon \in (0, 1)$, and $p = \Omega(\log(\frac{q}{\epsilon} \log N))$, there exists some $f \in \mathcal{A}'_{d,m,d',p}$ that $\epsilon$-approximates $q\mathrm{SA}$.*

*Proof.* Before explaining how they are produced by the input MLP, we introduce the corresponding key, value, and query inputs. The values will simply be $\phi(X)V = (z_1, \ldots, z_N)$. For some $m' = \frac{m-d}{2}$, let $\phi(X)K = (u_1, \ldots, u_N) \in \mathbb{R}^{N \times m'}$ be embedded key vectors, where $u_1, \ldots, u_N \in \{\pm 1/\sqrt{m'}\}^{m'}$ are the columns of a $m' \times N$ matrix satisfying the $(q, 1/4)$-restricted isometry and orthogonality property (Definition 5), as guaranteed to exist by Lemma 12 and the assumption on $m'$. Let $\alpha := \lceil 2 \log(4N/\epsilon) \rceil$. By Lemma 13, for each $y \in \binom{[N]}{q}$, there exists $w_y \in \mathbb{R}^{m'}$ with $\|w_y\|_2 \leq 2\sqrt{q}$ satisfying

$$\langle u_{i'}, w_y \rangle = 1 \quad \text{for all } i' \in y,$$

$$|\langle u_{i'}, w_y \rangle| \leq \frac{1}{2} \quad \text{for all } i' \notin y.$$

Given the bounded precision of the model, we are not free to represent the vectors $w_y$ exactly. Under $p$-bit precision for $p$ sufficiently large, we there exists a vector of $p$-bit floating point numbers $\widetilde{w_y} \in \mathbb{R}^{m'}$ for every $w_y$ with $\|w_y\|_2 \leq 2\sqrt{q}$ satisfying $\|\widetilde{w_y} - w_y\|_2 \leq \frac{\epsilon}{4\alpha}$. As an immediate consequence, $|\langle u_{i'}, \widetilde{w_y} \rangle - \langle u_{i'}, w_y \rangle| \leq \frac{\epsilon}{4\alpha}$ for all $i'$ and $y$ (by Cauchy-Schwarz). The remainder of the proof demonstrates that the necessary properties of the argument hold even with this approximation.

We now describe how to structure the neural network. We define an MLP $\phi : \mathbb{R}^d \to \mathbb{R}^m$ as $\phi(x_i) = \phi(z_i; y_i; i) = (z_i; \alpha \widetilde{w_{y_i}}; u_i)$, which works simply by using a look-up table on the values of $u_i$ and $\widetilde{w_{y_i}}$ from keys $i$ and $y_i$ respectively. Then, we define $Q, K, V$ as sparse boolean-valued matrices that simply copy their respective elements from $\phi(X)$.

We analyze the output of the softmax. If $i' \in y_i$, then

$$\mathrm{softmax}(\phi(X)QK^\top \phi(X)^\top)_{i,i'} = \frac{\exp(\alpha \langle u_i, \widetilde{w_{i'}} \rangle)}{\sum_{i'' \in y_i} \exp(\alpha \langle u_i, \widetilde{w_{i''}} \rangle) + \sum_{i'' \notin y_i} \exp(\alpha \langle u_i, \widetilde{w_{i''}} \rangle)}$$

$$\geq \frac{\exp(\alpha - \frac{\epsilon}{4})}{q \exp(\alpha + \frac{\epsilon}{4}) + N \exp(\frac{\alpha}{2} + \frac{\epsilon}{4})} = \frac{e^\alpha}{q e^\alpha + N e^{\alpha/2}} \cdot \exp\left(-\frac{\epsilon}{2}\right)$$

$$\geq \left(\frac{1}{q} - \frac{N e^{\alpha/2}}{q e^\alpha}\right)\left(1 - \frac{\epsilon}{2}\right) \geq \frac{\left(1 - \frac{\epsilon}{4}\right)\left(1 - \frac{\epsilon}{4}\right)}{q} \geq \frac{1}{q}\left(1 - \frac{\epsilon}{2}\right).$$

An analogous argument shows that

$$\mathrm{softmax}(\phi(X)QK^\top \phi(X)^\top)_{i,i'} \leq \frac{1}{q}\left(1 + \frac{\epsilon}{2}\right).$$

Likewise, if $i' \notin y_i$, then

$$\mathrm{softmax}(\phi(X)QK^\top \phi(X)^\top)_{i,i'} \leq \frac{\exp(\frac{\alpha}{2} + \frac{\epsilon}{4})}{q \exp(\alpha - \frac{\epsilon}{4})} \leq \exp\left(-\frac{\alpha}{2} + \frac{\epsilon}{2}\right) \leq \frac{\epsilon}{2N}.$$

We thus conclude that that we meet the desired degree of approximation for such $m$:

$$\|f(X)_i - q\mathrm{SA}(X)_i\|_2 = \left\| \sum_{i' \in y_i} \left(\frac{1}{q} - \mathrm{softmax}(\phi(X)QK^\top \phi(X)^\top)_{i,i'}\right) z_{i'} \right\|_2$$

$$+ \left\| \sum_{i' \notin y_i} \left(\mathrm{softmax}(\phi(X)QK^\top \phi(X)^\top)_{i,i'}\right) z_{i'} \right\|_2$$

$$\leq q \cdot \frac{\epsilon}{2q} + (N - q) \cdot \frac{\epsilon}{2N} \leq \epsilon. \qquad \square$$

### B.1.1 Restricted isometry and orthogonality property

The proof relies on the restricted isometry and orthogonality property from the compressed sensing literature. For $v \in \mathbb{R}^N$, let $\text{supp}(v) = \{i \in [N] : v_i \neq 0\}$.

**Definition 5.** We say a matrix $U \in \mathbb{R}^{m \times N}$ satisfies the $(q, \delta)$-*restricted isometry and orthogonality property* if

$$\|Uv\|_2^2 \in [(1 - \delta)\|v\|_2^2, (1 + \delta)\|v\|_2^2] \quad \text{and} \quad |\langle Uv, Uv' \rangle| \leq \delta \|v\|_2 \|v'\|_2$$

for all vectors $v, v' \in \mathbb{R}^N$ with $|\text{supp}(v)| \leq q$, $|\text{supp}(v')| \leq 2q$, and $\text{supp}(v) \cap \text{supp}(v') = \emptyset$.

The first result shows the existence of a sign-valued matrix $U$ that satisfies the desired distance-preserving property.

**Lemma 12** (Consequence of Theorem 2.3 of Mendelson et al. [2007] and Lemma 1.2 of Candes and Tao [2005])**.** *There is an absolute constant $C > 0$ such that the following holds. Fix $\delta \in (0, 1/2)$ and $q \in \mathbb{N}$. Let $U$ denote a random $m \times N$ matrix of independent Rademacher random variables scaled by $1/\sqrt{m}$. If $m \geq C(q \log N)/\delta^2$, then with positive probability, $U$ satisfies the $(q, \delta)$-restricted isometry and orthogonality property.*

Sparse subsets of the columns of such a $U$ can then be linearly separated from all other columns.

**Lemma 13** (Consequence of Lemma 2.2 in Candes and Tao [2005])**.** *Fix $\delta \in (0, 1/2)$ and $q \in \mathbb{N}$. Let matrix $U = [u_1, \ldots, u_N] \in \mathbb{R}^{m \times N}$ satisfy the $(q, \delta)$-restricted isometry and orthogonality property. For every vector $v \in \{0, 1\}^N$ with $\text{supp}(v) \leq q$, there exists $w \in \mathbb{R}^m$ satisfying*

$$
\begin{aligned}
\|w\|_2 &\leq \sqrt{q}/(1 - 2\delta), & \\
\langle u_i, w \rangle &= 1 & \text{if } v_i = 1, \\
|\langle u_i, w \rangle| &\leq \delta/(1 - 2\delta) & \text{if } v_i = 0.
\end{aligned}
$$

### B.2 Proof of Theorem 3

**Theorem 3** (Infinite-precision)**.** *For fixed $N$, $m \geq \Omega(d' + q)$ and $\epsilon > 0$, there exists some $f \in \mathcal{A}'_{d,m,d'}$ that $\epsilon$-approximates $q\text{SA}$.*

The proof relies on the properties of *neighborly polytopes*, which we define.

**Definition 6** (Ziegler [2006])**.** A polytope $P$ is $q$-*neighborly* if every subset of $q' \leq q$ vertices forms a $(q' - 1)$-face.

We give a $q$-neighborly polytope below that we use for the construction. For vectors $v_1, \ldots, v_N \in \mathbb{R}^{m'}$, let $\text{Conv}(v_1, \ldots, v_N) = \{\sum_{i=1}^N \alpha_i v_i : \alpha \in [0, 1]^N, \sum_i \alpha_i = 1\}$ denote their convex hull.

**Fact 14** (Theorem 1 of Gale [1963])**.** *For $t \in \mathbb{R}$, let $\theta(t) = (t, \ldots, t^{m'}) \in \mathbb{R}^{m'}$. Then, for all distinct $t_1, \ldots, t_N \in \mathbb{R}$, the* cyclic polytope *$\text{Conv}(\theta(t_1), \ldots, \theta(t_N))$ is $\frac{m'}{2}$-neighborly.*

The proof of Theorem 3 is immediate from the aforementioned fact and the following lemma.

**Lemma 15.** *Suppose there exists $u_1, \ldots, u_N \in \mathbb{R}^{m'}$ such that $\text{Conv}(u_1, \ldots, u_N)$ is $q$-neighborly. Then, for any $\epsilon > 0$, there exists some $f \in \mathcal{A}'_{d,m,d'}$ with fixed key vectors $\phi(X)K = (u_1, \ldots, u_N)$ that $\epsilon$-approximates $q\text{SA}$.*

*Proof.* The construction employs a similar look-up table MLP $\phi$ to the one used in the proof of Theorem 2. We let the key and value embeddings be

$$\phi(X)K = ((u_1, 1), \ldots, (u_N, 1)) \in \mathbb{R}^{N \times (m'+1)}, \text{ and } \phi(X)V = (z_1, \ldots, z_N) \in \mathbb{R}^{N \times d}.$$

To set the query vectors, observe that for any face $F$ of a polytope $P$, there exists a hyperplane $H_F$ such that $F \subset H_F$ and $P \setminus F$ lies entirely on one side of $H_F$. Thus, for every $y \in \binom{[N]}{q}$, there exists $w'_y \in \mathbb{R}^{m'}$ and $b_y \in \mathbb{R}$ such that

$$
w_y'^{\top} u_i + b_y \begin{cases} = 1 & \text{if } i \in y, \\ < 1 & \text{otherwise.} \end{cases}
$$

For $\alpha > 0$, let $\phi(x_i)^\mathsf{T} Q = \alpha w_y = \alpha(w'_y, b_y)$.

We construct the MLP to satisfy $\phi(x_i) = (z_k; w_{y_i}; u_i; 1) \in \mathbb{R}^m$ for $m = 2m' + 2$ and set parameter weights accordingly. Following the softmax analysis of Theorem 3, a sufficiently large choice of $\alpha$ ensures that $\max_{i \in [N]} \|f(X)_i - qSA(X)_i\|_2 \leq \epsilon$. □

## B.3 Proof of Theorem 4

**Theorem 4.** *For any sufficiently large $q$, any $N \geq 2q + 1$, and any $d' \geq 1$, there exists a universal constant $c$ such that if $mp \leq cq$, then no $f \in \mathcal{T}^{1,1}_{d,m,d',p}$ exists that $\frac{1}{2q}$-approximates $qSA$.*

*Proof.* We first embed every instance of DISJ with $n = q$ into an instance of $qSA$ and prove that they correspond. We assume the existence of the a transformer $f \in \mathcal{T}^{1,1}_{d,m,d',p}$ that $\frac{1}{2q}$-approximates $qSA$ and implies the existence of an $O(mp)$-bit communication protocol that computes DISJ. An application of Fact 5 concludes the proof.

Consider an instance of DISJ with $a \in \{0,1\}^q$ and $b \in \{0,1\}^q$ known by Alice and Bob respectively. We design an instance $X = (z_i; y_i; i)_{i \in [N]}$ of $qSA$. For each $j \in [2q]$, let $y_{2q+1} = \{2i + a_i - 1 : i \in [q]\}$. Additionally, let

$$z_j = \begin{cases} e_1 & \text{if } j \text{ is odd and } b_{(j-1)/2} = 1, \\ -e_1 & \text{otherwise.} \end{cases}$$

All other inputs are set arbitrarily. Then,

$$qSA(X)_{2q+1} = \frac{1}{q} \left| \{j \in [2q] : j \in y_{2q+1}, j \text{ is odd, and } a_{(j-1)/2} = 1\} \right| e_1$$

$$- \frac{1}{q} \left| \{j \in [2q] : j \in y_{2q+1} \text{ and } (j \text{ is even or } a_{(j-1)/2} = 0)\} \right| e_1$$

$$= \frac{|\{i \in [q] : a_i b_i = 1\}| - |\{i \in [q] : a_i b_i = 0\}|}{q} e_1.$$

Hence, $qSA(X)_{2q+1} = -e_1$ if and only if $\text{DISJ}(a,b) = 0$.

It remains to show that this implies the existence of an efficient communication protocol that computes $\text{DISJ}(a,b)$. By the existence of $f$, there exist $Q, K, V : \mathbb{R}^d \to \mathbb{R}^m$ and $\psi : \mathbb{R}^m \to \mathbb{R}^{d'}$ such that

$$f(X)_{2q+1} = \psi \left( \frac{\sum_{i=1}^N \exp\left(Q(x_{2q+1})^\mathsf{T} K(x_i)\right) V(x_i)}{\sum_{i=1}^N \exp\left(Q(x_{2q+1})^\mathsf{T} K(x_i)\right)} \right).$$

The protocol is as follows:

1. From $a$, Alice determines $y_{2q+1}$ and then computes $Q(x_{2q+1}) \in \mathbb{R}^m$, which she sends to Bob. This transmission uses $O(mp)$ bits.

2. Bob determines $z_1, \ldots, z_{2q}$ from $b$. Using those and the information from Alice, he computes $f(X)_{2q+1}$. He returns 1 if and only if $f(X)^\mathsf{T}_{2q+1} e_1 \geq -1 + \frac{1}{q}$.

The protocol computes $\text{DISJ}(a,b)$ because $f$ is a $\frac{1}{2q}$-approximation of $qSA$. Because any such protocol requires sharing $\Omega(q)$ bits of information, we conclude that $mp \leq cq$ for some $c$. □

## B.4 Optimality of Theorem 3 under restricted architectures

While the near-optimality of the bounded-precision self-attention construction in Theorem 2 is assured by the communication complexity argument of Theorem 4, it is not immediately apparent whether Theorem 3 is similarly optimal among infinite-precision self-attention models. Theorem 16 proves that this is indeed the case for a restricted family of architectures that resembles *cross-attention* rather than self-attention.

**Theorem 16.** *For input $x_1, \ldots, x_N$ satisfying $x_i = (z_i; y_i; i)$, suppose $\phi(x_i)^\mathsf{T} Q = w(y_i, i)$, $\phi(x_i)^\mathsf{T} K = u(i)$, and $\phi(x_i)^\mathsf{T} V = z_i$. Then, for any $q < N$ and $m \leq q(1 - C \log_N q)$ for some universal $C$, there do not exist $w : \mathbb{R}^d \times [N] \to \mathbb{R}^m$ and $u : [N] \to \mathbb{R}^m$ such that the resulting self-attention unit $\frac{1}{2q}$-approximates $q\mathrm{SA}$.*

The architectural assumptions of this statement are strong. For each element $x_i = (z_i; y_i; i)$, its value embedding must reproduce its target $z_i$; its key embedding depends exclusively on the index $i$; and its query embedding only on the indices $y_i$ and $i$. Indeed this attention unit more closely resembles *cross-attention* rather than self-attention, in which the problem is formulated as two sequences $((z_1, 1), \ldots, (z_N, N))$ and $(y_1; 1), \ldots, (y_N; N)$ that are passed to the key and value inputs and the query inputs respectively. We leave open the problem of generalizing this result to include all infinite-precision cross-attention or self-attention architectures, but we note that the constructions in Theorems 2 and 3 can be implemented under such architectural assumptions.

The proof relies on a geometric argument about how the convex hull of fixed key embeddings $U = (u(1), \ldots, u(N))$ lacks neighborliness and hence cannot separate every size-$q$ subsets of values embeddings $z_1, \ldots, z_N$ from the other values.

*Proof.* It suffices to show that for any fixed key embedding $U$, there exists some $y_i$ and setting of $z_1, \ldots, z_N$ such that

$$\left\| (\mathrm{softmax}(w(X)U^\mathsf{T})Z)_i - \frac{1}{q} \sum_{i' \in y_i} z_{i'} \right\|_2 \geq \frac{1}{2q},$$

where $w(X) = (w(y_1, 1), \ldots, w(y_N, N)) \in \mathbb{R}^{N \times m}$ and $U = (u(1), \ldots, u(N)) \in \mathbb{R}^{N \times m}$.

By Fact 17, for some $y_1 \in \binom{[N]}{q}$, there are no $w$ and $\tau \in \mathbb{R}$ satisfying $w(y_1, 1)^\mathsf{T} u_{i'} \geq \tau$ if and only if $i' \in y_1$. Hence, for any fixed $w$, there exists $i_1 \in y_1$ and $i_2 \in [N] \setminus y_1$ such that $w(y_1, 1)^\mathsf{T} u_{i_2} > w(y_1, 1)^\mathsf{T} u_{i_1}$. Given the value embeddings $z_{i_1} = e_1, z_{i_2} = e_2$ and $z_i = e_3$ for all $i \notin \{i_1, i_2\}$, we have

$$\left\| (\mathrm{softmax}(w(X)U^\mathsf{T})Z)_1 - \frac{1}{q} \sum_{i' \in y_1} z_{i'} \right\|_2^2 \geq \left( \mathrm{softmax}(w(X)U^\mathsf{T})Z)_{1, i_1} - \frac{1}{q} \right)^2 + (\mathrm{softmax}(w(X)U^\mathsf{T})Z)_{1, i_2}^2$$

$$\geq \max \left( \left( \mathrm{softmax}(w(X)U^\mathsf{T})Z)_{1, i_1} - \frac{1}{q} \right)^2, \mathrm{softmax}(w(X)U^\mathsf{T})Z)_{1, i_1}^2 \right)$$

$$\geq \frac{1}{4q^2}. \qquad \square$$

**Fact 17.** *If $m' < q(1 - \log_N Cq)$, then the columns of any $U = (u_1, \ldots, u_N) \in \mathbb{R}^{N \times m'}$ can be partitioned into sets $U_1$ and $U_2$ with $|U_1| = q$ that are not linearly separable. Hence, $\mathrm{Conv}(u_1, \ldots, u_N)$ is not $q$-neighborly.*

*Proof.* By the Sauer-Shelah Lemma [Sauer, 1972, Shelah, 1972, Vapnik and Chervonenkis, 1968] and the fact that the VC dimension of $m'$-dimensional linear thresholds is $m' + 1$, the maximum number of partitions of the columns of $U$ that can be linearly separated is at most

$$\sum_{k=0}^{m'+1} \binom{N}{i} \leq C' N^{m'+1} < C' \cdot \frac{N^q}{(Cq)^q} \leq \binom{N}{q},$$

for a sufficiently large choice of $C$ given universal constant $C'$. If the fact were to be false, then at least $\binom{N}{q} \geq (\frac{N}{q})^q$ such partitions must exist, which contradicts the above bound. $\qquad \square$

## C  Supplementary results for Section 4

### C.1  Proof of Theorem 6

**Theorem 6.** *For any input size $N$, input range $M = N^{O(1)}$, and fixed-precision bit complexity $p = O(\log M)$, there exists a transformer architecture $f \in \mathcal{T}_{1,m,1,p}^{1,1}$ with a single self-attention unit with embedding dimension $m = 3$ such that for all $X \in [M]^N$, $f(X) = \mathrm{Match2}(X)$.*

*Proof.* As discussed in Section 2.1, we allow a single blank token to be appended to the end of the sequence $X$ and assume the existence of a positional encoding. That is, we consider input $X' = (x_1, \ldots, x_N, x')$ with $x_{i,0} = i$ and $x' = \vec{0}$ to be the input to the target attention model. We define input MLP $\phi : \mathbb{R} \to \mathbb{R}^3$ and parameterizations $Q, K, V \in \mathbb{R}^{3 \times 3}$ such that

$$Q^{\mathsf{T}}\phi(x_i) = c\left(\cos\left(\frac{2\pi x_i}{M}\right), \sin\left(\frac{2\pi x_i}{M}\right), 1\right),$$

$$K^{\mathsf{T}}\phi(x_i) = \left(\cos\left(\frac{2\pi x_i}{M}\right), -\sin\left(\frac{2\pi x_i}{M}\right), 0\right),$$

$V^{\mathsf{T}}\phi(x_i) = \vec{1}$, $Q^{\mathsf{T}}\phi(x') = \vec{0}$, $K^{\mathsf{T}}\phi(x') = e_3$, and $V^{\mathsf{T}}\phi(x') = \vec{0}$. By elementary trigonometric identities, the following is true about the corresponding inner products:

$$(Q^{\mathsf{T}}\phi(x_i))^{\mathsf{T}}K^{\mathsf{T}}\phi(x_j) = c\cos\left(\frac{2\pi(x_i + x_j)}{M}\right)$$

$$(Q^{\mathsf{T}}\phi(x_i))^{\mathsf{T}}K^{\mathsf{T}}\phi(x') = cd.$$

As a result, $(Q^{\mathsf{T}}\phi(x_i))^{\mathsf{T}}K^{\mathsf{T}}\phi(x_j) = cd$ if and only if $x_i + x_j = 0 \,(\mathrm{mod}\, M)$. Otherwise, $(Q^{\mathsf{T}}\phi(x_i))^{\mathsf{T}}K^{\mathsf{T}}\phi(x_j) \le c(1 - \frac{1}{M^2})$. (Here, the $O(\log M)$-bit fixed-precision arithmetic is sufficient to numerically distinguish the two cases.) For each $i \in [N]$ let

$$\beta_i = |\{j \in [N] : x_i + x_j = 0 \,(\mathrm{mod}\, M)\}|$$

represent the total number of matches the input belongs to. If we take $c = M^2 \log(6N)$, then

$$(\mathrm{softmax}(\phi(X)QK^{\mathsf{T}}\phi(X)^{\mathsf{T}}))_{i,j} \in \begin{cases} [0, \frac{1}{6N}] & \text{if } x_i + x_j \neq 0 \,(\mathrm{mod}\, M) \text{ and } i, j \in [N]; \\ [\frac{1}{\beta_i + 1} \pm \frac{1}{6N}] & \text{if } x_i + x_j = 0 \,(\mathrm{mod}\, M) \text{ and } i, j \in [N]; \\ [\frac{1}{\beta_i + 1} \pm \frac{1}{6N}] & \text{if } i \in [N], j = N + 1. \end{cases}$$

We conclude that for any $i \in [N]$,

$$(\mathrm{softmax}(\phi(X)QK^{\mathsf{T}}\phi(X)^{\mathsf{T}})V\phi(X))_i \begin{cases} \le \frac{1}{6} \cdot \vec{1} & \text{if } \nexists j \text{ s.t. } x_i + x_j = 0 \,(\mathrm{mod}\, M) \\ \ge \left(\frac{\beta_i}{\beta_i + 1} - \frac{1}{6}\right) \cdot \vec{1} & \text{if } \exists j \text{ s.t. } x_i + x_j = 0 \,(\mathrm{mod}\, M), \end{cases}$$

where $\le$ is a partial ordering with $v \le v'$ if $v_i \le v_i'$ for all $i$. Since the latter case holds only when $\beta_i \ge 1$, the final step of the proof is design an output MLP $\psi$ such that $\psi(z) = 1$ if $z \ge \frac{1}{3}$ and $\psi(z) = 0$ if $z \le \frac{1}{6}$, which can be crafted using two ReLU gates. $\qquad\square$

### C.2  Proof of Theorem 7

**Theorem 7.** *There is universal constant $c > 0$ such that for sufficiently large $N$, and any $M \ge N + 1$, if $mpH \le cN/\log\log N$, then there is no $f \in \mathcal{T}_{1,m,1,p}^{1,H}$ satisfying $f(X) = \mathrm{Match3}(X)$ for all $X \in [M]^N$.*

*Proof.* The proof relies on a reduction to Fact 5 that embeds inputs to the set-disjointness problem of cardinality $n = \frac{N-1}{2}$ into a subset of instances passed to $\mathrm{Match3}$. For the sake of simplicity, we assume in the construction that $N$ is odd; if it were not, we could replace it with $N - 1$ and set the final element such that it never belongs to a triple.

We consider the following family of inputs to $\mathrm{Match3}$:

$$x_i \in \begin{cases} \{0\} & \text{if } i = 1, \\ \{1, i\} & \text{if } i \in \{2, \ldots, \frac{N+1}{2}\}, \\ \{1, (M - i + \frac{N-1}{2})\} & \text{if } i \in \{\frac{N+3}{2}, \ldots, N\}. \end{cases} \tag{3}$$

Note that $\mathrm{Match3}(X)_1 = 1$ if and only if there exists $i \in \{2, \ldots, \frac{N+1}{2}\}$ such that $x_i = i$ and $x_{i+\frac{N-1}{2}} = (M - i)$. Given input $(a, b) \in \{0, 1\}^n \times \{0, 1\}^n$ to DISJ, let $x_{i+1} = 1$ if and only if $a_i = 0$, and let $x_{i+\frac{N+1}{2}} = 1$ if and only if $b_i = 0$. Then, $\mathrm{Match3}(X)_1 = 1$ iff $\mathrm{DISJ}(a, b) = 1$.

Suppose $f(X) = \mathrm{Match3}(X)$ for all $X \in [M]^N$ for some $f \in \mathcal{T}_{1,m,1,p}^{1,H}$. We show that $f$ simulates an $O(mpH)$-bit communication protocol for testing DISJ. By definition of the standard self-attention unit with multi-layer perceptrons, note that $f(X)_1 = \psi(\sum_{h=1}^{H} f_h(\phi(X)))$ for $\phi : \mathbb{R} \to \mathbb{R}^m$, $\psi : \mathbb{R}^m \to \{0, 1\}$, and

$$f_h(X) = \frac{\sum_{i=1}^{N} \exp(Q_h(x_1)^\top K_h(x_i)) V_h(x_i)}{\sum_{i=1}^{N} \exp(Q_h(x_1)^\top K_h(x_i))},$$

for $Q_h, K_h, V_h : \mathbb{R}^{m \times m}$.

If we assume that this construction exists and is known explicitly by both Alice and Bob, we design a communication protocol for Alice and Bob to solve DISJ by sharing $O(mpH)$ bits with one another. Let Alice possess $a \in \{0, 1\}^n$ and Bob $b \in \{0, 1\}^n$, with $n = \frac{N-1}{2}$.

1. Alice and Bob compute $(x_2, \ldots, x_{\frac{N+1}{2}})$ and $(x_{\frac{N+3}{2}}, \ldots, x_N)$ from $a$ and $b$ respectively.

2. Alice computes an $O(p \log \log N)$-bit approximation of the logarithm of the first half of the softmax normalization term for each attention head and sends the result to Bob. That is, she sends Bob

$$L_{h,a} = \log \left( \sum_{i=1}^{\frac{N+1}{2}} \exp(Q_h(\phi(x_1))^\top K_h(\phi(x_i))) \right)$$

for each $h \in [H]$. This requires transmitting $O(pH \log \log N)$ bits.

3. Bob finishes the computation of normalization terms

$$L_h = \log \left( \exp(L_{h,a}) + \sum_{i=\frac{N+3}{2}}^{N} \exp(Q_h(\phi(x_1))^\top K_h(\phi(x_i))) \right)$$

for each $h$ and sends the result back to Alice (up to $O(p \log \log N)$-bits of precision). This again requires transmitting $O(pH \log \log N)$ bits.

4. Alice computes the partial convex combination of the first $\frac{N+1}{2}$ value vectors stipulated by the attention matrix

$$S_{h,a} = \frac{\sum_{i=1}^{\frac{N+1}{2}} \exp(Q_h(\phi(x_1))^\top K_h(\phi(x_i))) V_h(\phi(x_i))}{\exp(L_h)} \in \mathbb{R}^m$$

for each $h$ and sends the partial combinations to Bob. This requires transmitting $O(mpH \log \log N)$ bits (using the same precision as above).

5. Bob finishes the computation of the convex combinations

$$f_h(X) = S_{h,a} + \frac{\sum_{i=\frac{N+3}{2}}^{N} \exp(Q_h(\phi(x_1))^\top K_h(\phi(x_i))) V_h(\phi(x_i))}{\exp(L_h)} \in \mathbb{R}^m.$$

Bob concludes the protocol by computing and outputting $f(X)_1$, using his knowledge of each $f_h(X)$ and of $\psi$.

By the equivalences previously established, Bob returns 1 if and only if $\mathrm{DISJ}(a, b) = 1$. Because the protocol requires $O(mpH \log \log N)$ bits of communication, we can only avoid contradicting Fact 5 if $mpH \geq \Omega(n/\log \log N) = \Omega(N/\log \log N)$. □

**Remark 1.** *The domain restrictions to* $\mathrm{Match3}$ *stipulated in Equation* (3) *make the* $\mathrm{Match3}$ *problem substantially easier to solve than the full-domain case. Indeed, under the domain restrictions,*

$$\mathrm{Match3}(X)_1 = \max_{i \in \{2, \ldots, \frac{N+1}{2}\}} \mathrm{Match2}(X)_i,$$

*which is computable by a two-layer single-headed transformer network with constant embedding dimension. The first layer computes each* $\mathrm{Match2}(X)_i$ *with the construction in the proof of Theorem 6, and the second computes the maximum of the previous outputs by using those outputs as key vectors.*

*While Informal Conjecture 1 suggests that two layers are insufficient to compute the full-domain version of* $\mathrm{Match3}$*, this restricted variant introduces a concise* depth separation *(see Eldan and Shamir [2016], Telgarsky [2016], Daniely [2017]) between one- and two-layer transformer models.*

### C.3  Higher-order tensor attention

We introduce a novel category of higher-order tensor-based transformer models in order to show that problems like $\mathrm{Match3}$ that are hard to compute with standard transformer models can be made solvable. An $s$-order transformer is designed to efficiently compute dense $s$-wise interactions among input elements in an analogous manner to how standard transformers compute pairwise interactions. (We think of a standard transformer as second-order.) Before defining the new type of attention, we introduce notation to express the needed tensor products.

For vectors $v^1 \in \mathbb{R}^{N_1}$ and $v^2 \in \mathbb{R}^{N_2}$, let $v^1 \otimes v^2 \in \mathbb{R}^{N_1 N_2}$ denote their *Kronecker product* by $(v^1 \otimes v^2)_{(i_1-1)N_2+i_2} = v^1_{i_1} v^2_{i_2}$. The *column-wise Kronecker product* of matrices $A^1 \in \mathbb{R}^{N_1 \times m}$ and $A^2 \in \mathbb{R}^{N_2 \times m}$ is

$$A^1 \star A^2 = [A^1_1 \mid \cdots \mid A^1_m] \star [A^2_1 \mid \cdots \mid A^2_m] = [A^1_1 \otimes A^2_1 \mid \cdots \mid A^1_m \otimes A^2_m] \in \mathbb{R}^{N_1 N_2 \times m}.$$

The following generalizes the definition of self-attention.

**Definition 7.** For order $s \geq 2$, input dimension $d$, output dimension $d'$, embedding dimension $m$, bit complexity $p$, and matrices $Q, K^1, \ldots, K^{s-1} \in \mathbb{R}^{d \times m}$ and $V^1, \ldots, V^{s-1} \in \mathbb{R}^{d \times d'}$ (encoded with $p$-bit fixed-point numbers), an *$s$-order self-attention unit* is a function $f_{Q,K,V} : \mathbb{R}^{N \times d} \to \mathbb{R}^{N \times d'}$ with

$$f_{Q,K,V}(X) = \mathrm{softmax}(\underbrace{XQ}_{\in \mathbb{R}^{N \times m}} \underbrace{((XK^1) \star \cdots \star (XK^{s-1}))^{\intercal}}_{\in \mathbb{R}^{m \times N^{s-1}}}) \underbrace{((XV^1) \star \cdots \star (XV^{s-1}))}_{\in \mathbb{R}^{N^{s-1} \times d'}}.$$

The input to the row-wise softmax is an $N \times N^{s-1}$ matrix. Let $\mathcal{A}^{\otimes s}_{d,m,d',p}$ denote the set containing all such attention units.

Note that $\mathcal{A}^{\otimes 2}_{d,m,d',p} = \mathcal{A}_{d,m,d',p}$. Because $s$-order self-attention units have the same domain and codomain as standard self-attention, multiple units can be analogous combined to construct multi-headed attention units and full transformer models. We define $\mathcal{A}^{M,\otimes s}_{d,m,d',p}$ and $\mathcal{T}^{D,H,\otimes s}_{d,m,d',p}$ accordingly.

The purpose of the $s$-order transformer model as a theoretical construct is to posit how strictly generalizing the architecture in order to permit higher order outer products transfers the expressive powers of standard transformer architectures to more sophisticated interactions among elements of the input sequence $X$. The model is not defined to be immediately practical, due to its steep computational cost of evaluation.

However, the trade-offs involved in using such architectures resemble those already made by using transformer models instead of fully-connected networks. Transformers are already computationally wasteful relative to the number of the parameters, and these models likely succeed only because extremely efficient factorized parameterization exist. Likewise, third-order transformers could indeed be practical if even more factorization proves useful, since the computational costs may prove mild if the embedding dimension $m$, number of heads $H$, and depth $D$ necessary to succeed on a task exceed the sequence length $N$ for standard second-order transformers.

### C.4  Efficient representation of $\mathrm{Match3}$ with third-order self-attention

**Theorem 18** ($\mathrm{Match3}$ construction with third-order self-attention)**.** *For any sequence length $N$, input range $M = N^{O(1)}$, and fixed-precision bit complexity $p = O(\log M)$, there exists a third-order transformer architecture $f \in \mathcal{T}^{1,1,\otimes 3}_{1,m,1,p}$ with a single self-attention unit with embedding dimension $m = 5$ such that for all $X \in [M]^N$, $f(X) = \mathrm{Match3}(X)$.*

*Proof of Theorem 18.* The proof is almost identical to that of Theorem 6, except that we instead use a different key and query transforms to express a different trigonometric function:

$$Q\phi(x_i) = c\left(\cos\left(\frac{2\pi x_i}{M}\right), -\cos\left(\frac{2\pi x_i}{M}\right), \sin\left(\frac{2\pi x_i}{M}\right), \sin\left(\frac{2\pi x_i}{M}\right), 1\right),$$

$$K^1\phi(x_i) = \left(\cos\left(\frac{2\pi x_i}{M}\right), \sin\left(\frac{2\pi x_i}{M}\right), -\cos\left(\frac{2\pi x_i}{M}\right), \sin\left(\frac{2\pi x_i}{M}\right), 0\right),$$

$$K^2\phi(x_i) = \left(\cos\left(\frac{2\pi x_i}{M}\right), \sin\left(\frac{2\pi x_i}{M}\right), \sin\left(\frac{2\pi x_i}{M}\right), -\cos\left(\frac{2\pi x_i}{M}\right), 0\right).$$

Together, these ensure that the resulting tensor products reduce to a trigonometric expression that is maximized when $x_i + x_{j_1} + x_{j_2} = 0 \pmod{M}$. That is,

$$(\phi(X)Q((\phi(X)K^1) \star (\phi(X)K^2))^\top)_{i,(j_1-1)+j_2} = c\cos\left(\frac{2\pi(x_i + x_{j_1} + x_{j_2})}{M}\right).$$

We similarly let $V^1\phi(x_i) = V^2\phi(x_i) = \vec{1}$ and $V^1\phi(x') = V^2\phi(x') = \vec{0}$. The remaining choice of $c$ and the output MLP, and the analysis of the softmax proceeds identically to the previous proof. $\square$

### C.5 Heuristic argument for Informal Conjecture 1

**Conjecture 19** (Formal version of Informal Conjecture 1). *For sufficiently large $N$ and any $d \geq 1$, for all $M \geq N + 1$ and $mpHD \leq N^{\Omega(1)}$, there is no $f \in \mathcal{T}_{1,m,1,p}^{D,H}$ satisfying $f(X) = \text{Match3}(X)$ for all $X \in [M]^N$.*

We believe that the conjecture holds due to a heuristic information-theoretic argument. Define the distribution $\mathcal{D}$ over inputs $X \in \mathbb{R}^N$ that will be used to show that the model cannot compute Match3 for $M = N^4$ with high probability. We draw $\mathbf{X}$ from $\mathcal{D}$ as follows:

($E_1$) With probability $\frac{1}{2}$, draw each $\mathbf{x}_i$ iid from $\text{Unif}([M])$.

($E_2$) With probability $\frac{1}{2}$, draw $j_1, j_2, j_3$ iid from $\text{Unif}(\binom{[N]}{3})$. For all $i \neq j_3$, draw each $\mathbf{x}_i$ iid from $\text{Unif}([M])$. Let $\mathbf{x}_{j_3} = -\mathbf{x}_{j_1} - \mathbf{x}_{j_2} \pmod{M}$.

Note that under event $E_1$, a three matching elements exist with probability at most $\frac{1}{N}$, and

$$\Pr\left[\text{Match3}(\mathbf{X}) = \vec{0} \mid E_1\right] \geq 1 - \frac{1}{N}.$$

Under event $E_2$, a triple of matching elements is always planted, so $\text{Match3}(\mathbf{X}) \neq \vec{0}$. It would suffice to prove that—unless a transformer is sufficiently large—it is impossible to determine whether $\text{Match3}(\mathbf{X}) = \vec{0}$ with probability at least 0.9.

Under $\mathcal{D}$, any subset of $\{\mathbf{x}_1, \ldots, \mathbf{x}_N\}$ consists of iid integers drawn uniformly from $[M]$, unless all of $\mathbf{x}_{j_1}, \mathbf{x}_{j_2}, \mathbf{x}_{j_3}$ appear in the subset. Consider a transformer architecture with $p$-bit precision, $m$-dimensional embeddings, $H$ heads per layer, and $D$ layers. We argue informally that a single-element output of a self-attention unit can take into account information about $mp$ more inputs $\mathbf{x}_1, \ldots, \mathbf{x}_N$ than that it had in the previous layer. By induction, after $D$ layers of $H$-headed self-attention with interleaved MLPs, each element is a function of at most $mpHD$ inputs. Until an element exists that is a function of at least two of the three of $\mathbf{x}_{j_1}, \mathbf{x}_{j_2}, \mathbf{x}_{j_3}$, we assume that the elements "known" by each output are chosen independently of the indices $j_1, j_2, j_3$. (Given two elements of the triple, the third element can be identified with a single self-attention unit.) Hence, we argue that it suffices to show that the probability any two elements of the triple $j_1, j_2, j_3$ occurring within any of the $N$ sets of $mpHD$ inputs is vanishingly small for sufficiently large transformer parameters. The probability of single collection having any of two of the three inputs is at most

$$\frac{3\binom{mpHD}{2}}{\binom{N}{2}} \leq 3\left(\frac{empHD}{N}\right)^2.$$

Thus, the probability that any collection has all three inputs is no more than $3(empHD)^2/N$. If $mpHD = O(\sqrt{N})$, then the randomly chosen triple will not jointly appear as the outcome of a single

element of a self-attention unit with probability at least $0.9$, and the transformer will be unexpected to successfully distinguish between the two cases.

Should the conjecture hold, it would represented a tight lower bound on the size of the smallest standard transformer architecture necessary to compute $\mathrm{Match3}$.

**Theorem 20** (Tightness of Conjecture 19). *For any sequence length $N$, if the input range satisfies $M = N^{O(1)}$ and the transformer size parameters satisfy $p \geq \log(M)$, $H = 1$, $m \geq 4$, and $mD \geq CN^2$ for some universal constant $C$, then there exists a transformer architecture $f \in \mathcal{T}_{1,m,1,p}^{D,H}$ such that $f(X) = \mathrm{Match3}(X)$.*

*Proof.* We construct an architecture that collects a group of candidate pairs in each layer of single-headed self-attention and verifies whether there exists a triple incorporating each pair that satisfies the summation property. Then, all candidate triples are disposed of, and the subsequent layer collects a new family of candidates.

To do so, we first let $\ell := \lfloor \frac{m}{2} \rfloor - 1 \geq 1$ represent the total number of pairs shared in each layer of attention. We let $P = \binom{[N]}{2}$ represent a collection of all pairs of indices and partition it into $D$ subsets $P_1, \ldots, P_D$, each containing $\ell$ distinct pairs. (Since $|P| = \frac{N(N+1)}{2}$, any $D$ satisfying the theorem's preconditions is sufficiently large for this to be a proper partition.) Our construction ensures that there exist $x_i + x_{j_1} + x_{j_2} = 0 \pmod{M}$ for $(j_1, j_2) \in P_k$, then the $k$th layer of self attention will verify its existence and mark $x_i$ as belonging to the match. Throughout the network, we maintain that the first two dimensions of any embedding of the $i$th element correspond to $x_i \in [M]$ and a bit indicating whether a match has been found yet containing $x_i$.

Consider the first layer of self-attention, and let $P_1 = \{(i_1, j_1), \ldots, (i_\ell, j_\ell)\}$. We set the input MLP $\phi_1 : \mathbb{R}^d \to \mathbb{R}^m$ and respective matrices $Q^1, K^1 \in \mathbb{R}^{m \times m}$ such that

$$Q^1 \phi_1(x_i) = c e_1 \text{ and } K^1 \phi_1(x_i) = \begin{cases} e_1 & \text{if } i \in P_1 \\ \vec{0} & \text{otherwise,} \end{cases}$$

for sufficiently large $c$. We additionally let

$$V^1 \phi_1(x_i) = \begin{cases} (2\ell + 1) \cdot (x_i; 0; \vec{0}) & i \notin P_1, \\ (2\ell + 1) \cdot (x_i; 0; x_i e_{2\iota - 1}) & i = i_\iota, \\ (2\ell + 1) \cdot (x_i; 0; x_i e_{2\iota}) & i = j_\iota. \end{cases}$$

By making use of a residual connection, we ensure that the $i$th outcome of the self-attention is $(x_i, 0, x_{i_1}, x_{j_1}, \ldots, x_{i_\ell}, x_{j_\ell})$. We encode an MLP to compute

$$(x_i, 0, x_{i_1}, x_{j_1}, \ldots, x_{i_\ell}, x_{j_\ell}) \mapsto \left( x_i, \mathbb{1}\left\{ \exists \iota \in [\ell] \text{ s.t. } x_i + x_{i_\iota} + x_{j_\iota} = \vec{0} \pmod{M} \right\}; \vec{0} \right).$$

We repeat this construction $D$ times, with the only modifications being the replacement of $P_1$ and the fact that the second dimension of the embedding remains 1 after being set to that value. After $D$ layers, the final MLP outputs the value of the second dimension, which will be 1 if and only if the respective $x_i$ belongs to a three-way match. □

## C.6 Sharper separations for embedded subgraph detection problems

In pursuit of proving separations analogous to the one between Theorem 18 and Conjecture 19, we draw techniques for proving lower bounds for graph problems in the CONGEST model of distributed computation with restricted bandwidth [Peleg, 2000].[5]

---

[5]At a high level, the CONGEST model features $N$ players that communicate in synchronous rounds over a network (an undirected graph with $[N]$ as its vertices) to solve a computational problem Peleg [2000]. In each round, each player can send a message to each of its neighbors. The computation that each player does with the messages received from its neighbors is unrestricted; the primary resources considered in CONGEST is the number of rounds of communication and the message sizes. Although CONGEST is often studied for solving computational problems on input graphs with vertices $[N]$, the input graph need not be the same as the communication network.

The problems we consider take, as input, the adjacency matrix $X \in \{0, 1\}^{N \times N}$ of an $N$-vertex graph $G = (\mathcal{V}, \mathcal{E})$ with $\mathcal{V} = [N]$, so $x_{i,j} = \mathbb{1}\{(i, j) \in \mathcal{E}\}$. We may regard each row of $X$ as a high-dimensional $(d = N)$ embedding of the $i$-th vertex containing information about which (outgoing) edges are incident to the $i$-th vertex. We consider the following problems:

$$\mathrm{DirectedCycle3}(X) = (\mathbb{1}\{\exists j_1, j_2 \in [N] \text{ s.t. } x_{i,j_1} x_{j_1,j_2} x_{j_2,i} = 1\})_{i \in [N]};$$

$$\mathrm{Cycle5}(X) = (\mathbb{1}\{\exists j_1, j_2, j_3, j_4 \in [N] \text{ s.t. } x_{i,j_1} x_{j_1,j_2} x_{j_2,j_3} x_{j_3,j_4} x_{j_4,i} = 1\})_{i \in [N]},$$

$$\text{with } \mathrm{dom}(\mathrm{Cycle5}) = \{X : X = X^\top\}.$$

The former treats $X$ as a directed graph (where $X$ need not be symmetric) and asks whether each input belongs to a directed 3-cycle. The latter insists that $X$ be an undirected graph by enforcing symmetry and determines membership in (undirected) 5-cycles.

However, solving these problems with any transformer model of constant order trivially requires having the product of the precision $p$, embedding dimension $m$, heads per layer $H$, and depth $D$ grow polynomially with $N$, since each attention unit is limited to considering at most $pm$ bits of information from each input. Such a lower bound is not interesting for dense graphs, where every vertex may have $\Omega(N)$ incident edges; the bottleneck is not due to any feature of standard attention units (and would persist with higher-order attention).

To circumvent this issue, we consider an augmented self-attention unit, which permits each element of the self-attention tensor to depend on both its respective inner product and on the presence of edges among corresponding inputs.

**Definition 8.** For order $s \geq 2$, input dimension $d$, output dimension $d'$, embedding dimension $m$, bit complexity $p$, matrices $Q, K^1, \ldots, K^{s-1} \in \mathbb{R}^{d \times m}$ and $V^1, \ldots, V^{s-1} \in \mathbb{R}^{d \times d'}$ (encoded with $p$-bit fixed-point numbers), and cell-wise attention tensor function $\kappa : \{0, 1\}^{s(s-1)} \times \mathbb{R} \to \mathbb{R}$, an $s$-order graph self-attention unit is a function $f_{Q,K,V} : \mathbb{R}^{N \times d} \to \mathbb{R}^{N \times d'}$ with

$$f_{Q,K,V}(X) = \mathrm{softmax}(\kappa(X, XQ((XK^1) \star \cdots \star (XK^{s-1}))^\top))((XV^1) \star \cdots \star (XV^{s-1})).$$

For attention tensor $A \in \mathbb{R}^{N^{\otimes s}}$, we abuse notation by writing $\kappa(X, A)$ as short-hand for the particular cell-wise application of a fixed function, incorporating information about all relevant edges:

$$\kappa(X, A)_{i_1, \ldots, i_s} = \kappa(x_{i_1, i_2}, x_{i_1, i_3}, \ldots, x_{i_s, i_{s-1}}, x_{i_s, i_{s-2}}, A_{i_1, \ldots, i_s}).$$

Let $\mathcal{AG}^{\otimes s}_{d,m,d',p}$ and $\mathcal{TG}^{D,H,\otimes s}_{d,m,d',p}$ denote all such attention units and all such transformers respectively.

Now, we provide four results that exhibit separations between orders of graph self-attention.

**Theorem 21** (Hardness of representing Cycle5 with standard graph transformer). *For sufficiently large $N$, any $f \in \mathcal{TG}^{D,H}_{N,m,1,p}$ satisfying $f(X) = \mathrm{Cycle5}(X)$ for all $X \in \{0, 1\}^{N \times N}$ with $X = X^\top$ requires $mpHD = \Omega(N/\log^2 N)$.*

**Theorem 22** (Efficient construction of Cycle5 with fifth-order graph transformer). *For sequence length $N$ and bit-complexity $p = O(\log N)$, there exists a fourth-order graph transformer architecture $f \in \mathcal{TG}^{1,1,\otimes 5}_{N,1,1,p}$ with a single graph self-attention unit such that for all $X \in \{0, 1\}^{N \times N}$ with $X = X^\top$, $f(X) = \mathrm{Cycle5}(X)$.*

**Theorem 23** (Hardness of representing DirectedCycle3 with standard graph transformer). *For sufficiently large $N$, any $f \in \mathcal{TG}^{D,H}_{N,m,1,p}$ satisfying $f(X) = \mathrm{DirectedCycle3}(X)$ for all $X \in \{0, 1\}^{N \times N}$ requires $mpHD = \Omega(N/\log^2 N)$.*

**Theorem 24** (Efficient construction of DirectedCycle3 with fourth-order graph transformer). *For sequence length $N$ and bit-complexity $p = O(\log N)$, there exists a third-order graph transformer architecture $f \in \mathcal{TG}^{1,1,\otimes 3}_{N,1,1,p}$ with a single graph self-attention unit such that for all $X \in \{0, 1\}^{N \times N}$, $f(X) = \mathrm{DirectedCycle3}(X)$.*

The proofs of Theorems 22 and 24 are immediate from the construction. Because each cell of the self-attention tensor has explicit access the the existence of all relevant edges, $\kappa$ can be configured to ensure that cell's value is large if and only if the requisite edges for the desired structure all exist. Taking a softmax with a blank element (like in Theorem 6) ensures that the outcome of the

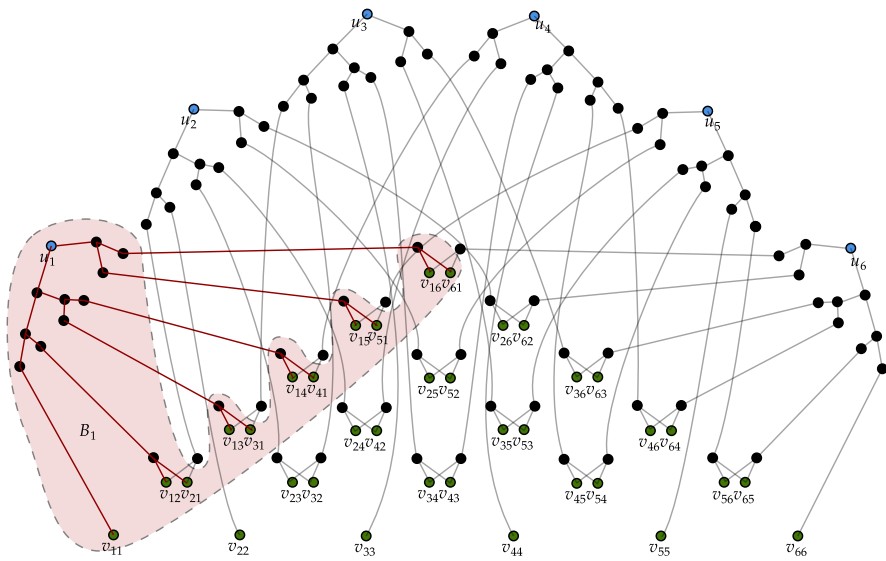

Figure 4: The CONGEST graph $G^N$ visualized for $N = 6$ with root nodes $\{u_i\}_{i \in [N]}$ in blue, leaf nodes $\{v_{i,j}\}_{i,j \in [N]}$ in green, and the nodes $V_1$ of the binary tree $B_1$ shaded red and edges $E_1$ colored red.

self-attention unit for a given element distinguishes between whether or not it belongs to a 5-cycle or a directed 3-cycle. The output MLP ensure that the proper output is returned.

We prove Theorems 21 and 23 by introducing a particular CONGEST communication graph that can be used to simulate any model in $\mathcal{T}\mathcal{G}_{d,m,d',p}^{D,H}$ (and hence, also any model in $\mathcal{T}_{d,m,d',p}^{D,H}$) in $O(mHD \log N)$ rounds of communication. Then, we show for each problem that we can encode each instance of the set disjointness communication problem as an instance of Cycle5 (or DirectedCycle3) and derive a contradiction from the communication graph.

### C.6.1 A CONGEST communication graph that generalizes standard graph transformer computation

The key principle of our analysis is that the predominant limitation of a transformer model is in its communication bandwidth and *not* its computational abilities. We model transformers as having element-wise multi-layer perceptron units with unbounded computational ability (but bounded precision inputs and outputs) and self-attention units, which compute linear combinations of inputs in a carefully regimented way that limits the ability of individual elements to share information with one another. Here, we introduce a specific CONGEST graph for each sequence length $N$ and show that every transformer has a communication protocol that simulates its computation in this graph.

For fixed $N$, we design an undirected CONGEST graph $G^N = (V^N, E^N)$ with $O(N^2)$ nodes, each having degree at most 3. (Note that this graph is *not* the same as the graph provided as input $X$ to a transformer; this graph is consistent across all transformers taking input of sequence size $N$.) Let $u_1, \ldots, u_N$ be nodes in $V^N$ corresponding to each input. For every pair $i, j \in [N]$, let $v_{i,j}$ be a node as well. For each $i \in [N]$, let $B_i = (V_i, E_i)$ be a balanced binary trees having root $u_i$ and leaves $v_{i,1}, \ldots, v_{i,N}, v_{1,i}, \ldots, v_{N,i}$. Hence, each $B_i$ has $O(N)$ vertices of degree 3 and is of depth $O(\log N)$. Let $V^N = V_1 \cup \cdots \cup V_N$ and $E^N = E_1 \cup \cdots \cup E_N$. Noting that $E_1, \ldots, E_N$ are disjoint and that $V_1, \ldots, V_N$ are disjoint, except for leaves $v_{i,j}$, we ascertain that $G^N$ contains $O(N^2)$ vertices of degree at most 3 and has diameter $O(\log N)$. We visualize the graph $G^N$ with a highlighted tree $B_1$ in Figure 4.

**Lemma 25.** *For any transformer $f \in \mathcal{T}\mathcal{G}_{d,m,d',p}^{D,H}$ and any $X \in \mathbb{R}^{N \times d}$ with $p$-bit fixed-precision numbers, there exists a CONGEST communication protocol on the graph $G^N$ that shares $p$ bits of information between adjacent vertices per round satisfying the following characteristics:*

- *Before any communication begins, each node $u_i$ is provided with $x_i$ and each node $v_{i,j}$ is provided with $x_{i,j}$ and $x_{j,i}$.*

- *After $T = O(HD(m + \log N))$ rounds of communication, each node $u_i$ outputs $f(X)_i$.*

*Proof.* It suffices to give a protocol that computes the outcome of a single-headed unit of graph self-attention with parameters $Q, K, V \in \mathbb{R}^{m \times m}$ and $\kappa : \{-1, 1\}^2 \times \mathbb{R} \to \mathbb{R}$ and transmits its $i$th output back to $u_i$ in $O(m \log N)$ rounds of $p$-bit communication. The remainder of the argument involves computing the outcomes of all element-wise MLPs within respective vertices $u_1, \ldots, u_N$ (since we assume each node to have unbounded computational power in the CONGEST model) and to repeat variants of the protocol $HD$ times for every individual self-attention unit. Because the protocol is designed for a particular transformer architecture $f$, we can assume that every node in the CONGEST graph has knows every parameter of $f$.

We give the protocol in stages. We assume inductively that every input to $f$, $y_1, \ldots, y_N \in \mathbb{R}^m$, is known by its respective vertex $u_1, \ldots, u_N$.

1. Every vertex $u_i$ computes $Q^\intercal y_i \in \mathbb{R}^m$ and propagates it to every vertex $v_{i,1}, \ldots, v_{i,N}$. This can be done in $O(m + \log N)$ rounds by transferring one $p$-bit fixed-precision number per round from an element of the binary tree $B_i$ to each of its children per round. Because the respective edges $E_1, \ldots, E_N$ are disjoint, this operation can be carried out in parallel.

2. Each $u_i$ computes $K^\intercal y_i, V^\intercal y_i \in \mathbb{R}^m$ and propages them to $v_{1,i}, \ldots, v_{N,i}$ in $O(m + \log N)$ rounds.

3. Each $v_{i,j}$, using their knowledge of $x_{i,j}$ and $x_{j,i}$, computes $\alpha_{i,j} := \exp(\kappa(x_{i,j}, x_{j,i}, y_i^\intercal Q K^\intercal y_j))$. This takes zero rounds.

4. Each $u_i$ computes $\sum_{j=1}^N \alpha_{i,j}$ by propagating each $\alpha_{i,j}$ in $v_{i,j}$ up $B_i$ to $u_i$, iteratively summing terms passed up. This takes $O(\log N)$ rounds.

5. Similarly, $u_i$ computes $\sum_{j=1}^N \alpha_{i,j} V^\intercal y_j$ in $O(m \log N)$ rounds. Then, it computes

$$\frac{\sum_{j=1}^N \alpha_{i,j} V^\intercal y_j}{\sum_{j=1}^N \alpha_{i,j}},$$

   which is the target output of the self-attention unit.

Because all steps are achievable in parallel with $O(m + \log N)$ rounds, the claim follows. $\square$

### C.6.2 Reduction from set disjointness

Before proving Theorems 21 and 23 by embedding an instance of a transformer model into an instance of each subgraph identification problem, we first introduce a partition of the vertices $V^N$ of the CONGEST graph into those possessed by Alice and Bob for use in a two-party communication protocol. We call those two sets $V_a^N$ and $V_b^N$.

Note that the previous section made no assumptions about the organization of edges in the binary tree. We thus add an additional condition: that each binary tree $B_i$ can be oriented to respect the left-to-right ordering $v_{i,1}, v_{1,i}, \ldots, v_{i,N}, v_{N,i}$. Let $u_i \in V_a^N$ if and only if $i \leq \frac{N}{2}$, and $v_{i,j} \in V_a^N$ if and only if $\min(i, j) \leq \frac{N}{2}$. We label are remaining nodes in $B_i$ by labeling a parent node $w_p$ as a function of its child nodes $w_\ell$ and $w_r$ using the following rules:

(a) If $w_\ell, w_r \in V_a^N$, then let $w_p \in V_a^N$.

(b) If $w_\ell, w_r \in V_b^N$, then let $w_p \in V_b^N$.

(c) Otherwise, let $w_p \in V_a^N$ if and only if root $u_i \in V_a^N$.

This partition, which we visualize in Figure 5, bounds the number of bits Alice and Bob can exchange by simulating a protocol on CONGEST graph $G^N$.

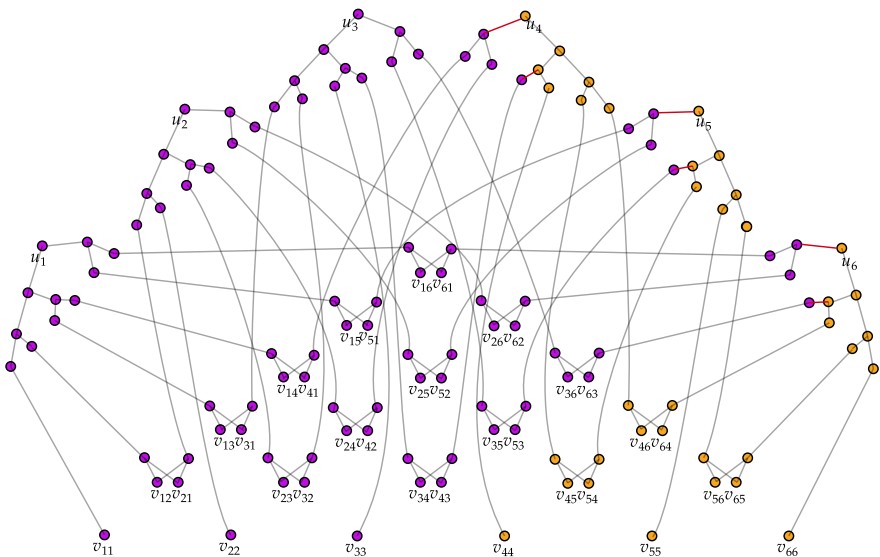

Figure 5: The CONGEST graph $G^N$ with vertices partitioned into sets $V_a^N$ (violet) and $V_b^N$ (orange) for $N = 6$. The six edges cut by the partition are colored red.

**Lemma 26.** *Suppose Alice and Bob simulate an $R$-round $p$-bit protocol on CONGEST communication graph $G^N$ where Alice has access to all vertices $V_a^N$ and Bob $V_b^N$. No other communication is permitted besides sharing bits as permitted by the CONGEST protocol between neighboring vertices. Then, Alice and Bob exchange at most $O(pRN \log N)$ bits.*

*Proof.* It suffices to show that the partition $V_a^N, V_b^N$ induces a cut of size at most $O(N \log N)$; this ensures that each can send no more than $O(pN \log N)$ bits per round.

Per the rules defined above, an edge in $(w_p, w_\ell)$ and $(w_p, w_r)$ is cut if and only if they are described by case (c). Within each tree $B_i$ under the orientation described above, an inductive argument shows that in every layer, all elements in $V_a^N$ are to the left of all elements in $V_b^N$. Thus, there exists at most one parent of that layer that belongs to case (c), and thus, no more than one cut edge per layer. Because each tree has $O(\log N)$ layers and because there are $N$ trees, the partition cuts at most $O(N \log N)$ edges. □

It remains to embed an instance of DISJ in $V_a^N, V_b^N$ for each problem such that its output corresponds identically with that of DISJ.

*Proof of Theorem 21.* Assume for the sake of simplicity that $N$ is divisible by 5. Let $a, b \in \{0,1\}^n$ for $n = \frac{N^2}{25}$ be an input to DISJ, and let Alice and Bob possess $a$ and $b$ respectively. We index those vectors as $a = (a_{1,1}, a_{1,2}, \ldots, a_{N/5,N/5-1}, a_{N/5,N/5})$ and $b = (b_{1,1}, \ldots, b_{N/5,N/5})$ for ease of analysis. We design input matrix $X \in \{0,1\}^{N \times N}$ as follows:

- If $i \in (0, \frac{N}{5}]$ and $j \in (\frac{N}{5}, \frac{2N}{5}]$, then $x_{i,j} = x_{j,i} = a_{i,j-N/5}$.

- If $i \in (\frac{N}{5}, \frac{3N}{5}]$ and $j \in (\frac{2N}{5}, \frac{4N}{5}]$, then $x_{i,j} = x_{j,i} = \delta_{i,j-N/5}$.

- If $i \in (\frac{3N}{5}, \frac{4N}{5}]$ and $j \in (\frac{4N}{5}, N]$, then $x_{i,j} = x_{j,i} = b_{j-4N/5,i-3N/5}$.

- If $i \in (\frac{4N}{5}, N]$ and $j \in (0, \frac{N}{5}]$, then $x_{i,j} = x_{j,i} = \delta_{i,j+4N/5}$.

- Otherwise, $x_{i,j} = 0$.

This ensures that $X$ has a 5-cycle if and only there exist $i, j \in (0, \frac{N}{5}]$ such that $a_{i,j} b_{i,j} = 1$[6]. In addition, note that under the protocol in Lemma 25, Alice's and Bob's inputs $a$ and $b$ are known exclusively by nodes belonging to $V_a^N$ and $V_b^N$ respectively.

Consider any transformer architecture $f \in \mathcal{TG}_{N,m,1,p}^{D,H}$ that computes Cycle5. By Lemma 25, there exists a protocol on the CONGEST graph $G^N$ that computes Cycle5 after $O(HD(m + \log N))$ rounds of communication of $p$-bits each. If Alice and Bob simulate this protocol, and output 1 if and only if at least one of their outputs indicates the existence of a Cycle5, then they successfully decide DISJ. By Lemma 26, this communication algorithm solves DISJ after exchanging $O(mpHDN \log^2 N)$ bits of communication. However, Fact 5 implies that no communication algorithm can do so without exchanging $\Omega(n) = \Omega(N^2)$ bits, which concludes the proof. □

*Proof of Theorem 23.* The proof is identical to its predecessor, but uses a different embedding of an instance $a, b \in \{0,1\}^n$ to DISJ. Let $n = \frac{N^2}{16}$. Then:

- If $i \in (0, \frac{N}{4}]$ and $j \in (\frac{N}{2}, \frac{3N}{4}]$, then $x_{i,j} = a_{i,j-N/2}$.

- If $i \in (\frac{N}{2}, \frac{3N}{4}]$ and $j \in (\frac{3N}{4}, N]$, then $x_{i,j} = b_{j-3N/4,i-N/2}$.

- If $i \in (\frac{3N}{4}, N]$ and $j \in (0, \frac{N}{4}]$, then $x_{i,j} = \delta_{i,j+3N/4}$.

- Otherwise, $x_{i,j} = 0$.

This construction ensures that a directed 3-cycle exists if and only if a corresponding pair of elements in $a$ and $b$ are both 1. □

# D   Experiment details

This section describes the experimental setup behind Figure 2, and provides further experiments suggesting an *implicit bias* of transformers for $q$SA, in particular when compared with MLPs and RNNs.

**Experimental setup.**   Experiments used synthetic data, generated for $q$SA with $n = 1000$ training and testing examples, a sequence length $N = 20$, $q = 3$, with the individual inputs described in more detail as follows.

- The positional encoding of element $i$ is a random vector sampled uniformly from the sphere in $\mathbb{R}^{d_0}$ with $d_0 := \lceil 1 + 2\ln(N) \rceil$, a quantity which agrees with the theory but was not tuned.

- A sequence element then consists of the data portion $z \in \mathbb{R}^{d_1}$ where $d_1 = 4$, also sampled from the unit sphere, then the positional encoding of this sequence element, and then $q$ further positional encodings identifying elements to average to produce the output; this differs from (and is more tractable than) the presentation in Section 3, where the positional encoding is provided as an integer and the MLP layer input to our attention layers is expected to choose a sufficient positional encoding.

As such, the total dimension of a sequence element is $d_1 + (q+1)d_0 = 32$. The architectures are detailed as follows.

- The attention is identical to the description in the paper body, with the additional detail of the width and embedding dimension $m$ being fixed to 100.

- Figure 6 also contains an MLP, which first flattens the input, then has a single hidden ReLU layer of width 256, before a final linear layer and an output reshaping to match the desired output sequence shapes.

- Figure 6 also contains an LSTM, which is a standard `pytorch` LSTM with 2 layers and a hidden state size 800, which is 200 times larger than the target output dimension 4.

---

[6]We consider 5-cycles rather than 4-cycles because a spurious 4-cycle could exist among edges $\{x_{i,j} : i \in (0, \frac{N}{5}], j \in (\frac{N}{5}, \frac{2N}{5}]\}$.

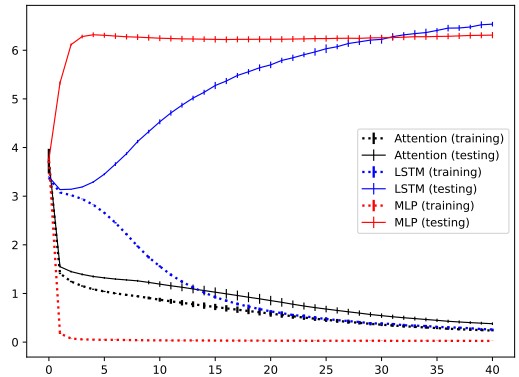

Figure 6: Test and train error curves of fitting various architectures to $q$SA, where the horizontal axis denotes thousands of training iterations, and the vertical axis denotes the regression objective; see Section D for further details.

Experiments fit the regression loss using Adam and a minibatch size of 32, with default precision, and take a few minutes to run on an NVIDIA TITAN XP, and would be much faster on standard modern hardware.

**Further discussion of Figure 2 and Figure 7.** In Figure 2 and Figure 7, we plot (post-softmax) alignment matrices after $T \in \{0, 1000, 40000\}$ iterations of Adam. The alignment matrices in Figure 2 are taken from the training example whose loss is the median loss across all examples. Figure 7 is similar, but additionally shows the examples of minimal and maximal loss.

**Further discussion of Figure 6.** Figure 6 plots training and testing error curves for the same attention architecture as in Figure 2, but with further MLP and LSTM architectures as described above. but also an MLP trained on flattened (vectorized) error bars reflect 5 separate training runs from random initialization. A few variations of these architectures were attempted, however curves did not qualitatively change, and in particular, only the attention layer achieves good generalization across all attempts.

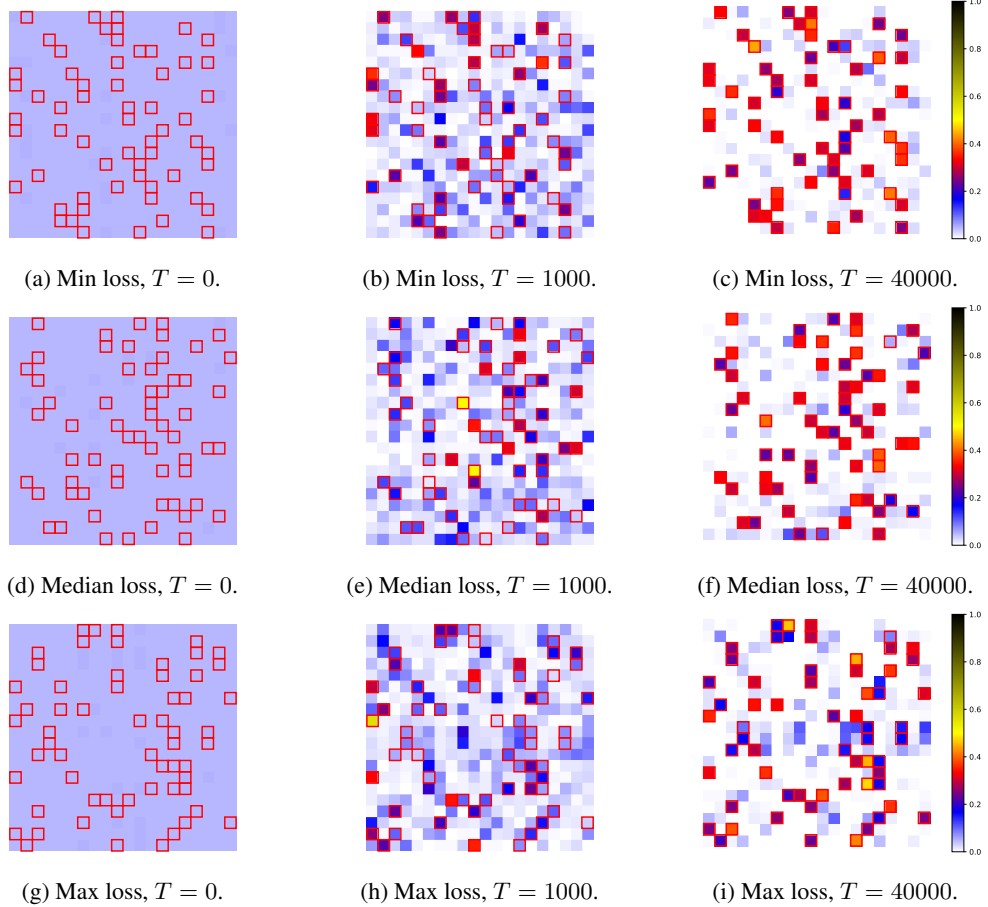

(a) Min loss, $T = 0$.  (b) Min loss, $T = 1000$.  (c) Min loss, $T = 40000$.

(d) Median loss, $T = 0$.  (e) Median loss, $T = 1000$.  (f) Median loss, $T = 40000$.

(g) Max loss, $T = 0$.  (h) Max loss, $T = 1000$.  (i) Max loss, $T = 40000$.

Figure 7: Alignment plots as in Figure 2, but using examples with minimum, median, and maximum loss, whereas Figure 2 only uses the example with median loss.

