$\mathrm{rank}(W) \le \frac{Nd'}{2} - 1$, then so too is $\mathrm{rank}([V_q; \ldots; V_N]) \le \frac{Nd'}{2} - 1$, and $[V_q; \ldots; V_N]$ has a nontrivial null space containing a nonzero vector $u = (u_q; \ldots; u_N) \in \mathbb{R}^{(N-q)d'}$. Let

$$\xi = \frac{1}{\max_{j \in \{q, \ldots, N\}} \|u_j\|_2} (u_q; \ldots; u_N),$$

$z = (\vec{0}; \ldots; \vec{0}; \xi_q; \ldots; \xi_N)$, and $z' = (\vec{0}; \ldots; \vec{0}; -\xi_q; \ldots; -\xi_N)$. Then,

1. $z_j, z'_j \in \mathbb{B}^{d'}$ for all $j \in [N]$;

2. $V_j z_j = V_j z'_j = 0$ for all $j \in [N]$; and

3. $\|z_{j^*} - z'_{j^*}\|_2 = 2$ for some $j^* \in \{q, \ldots, N\}$.

Therefore, for any $y_1, \ldots, y_N \in \binom{[N]}{q}$, respective $x = (1; \ldots; N; y_1; \ldots; y_N; z_1; \ldots; z_N)$ and $x' = (1; \ldots; N; y_1; \ldots; y_N; z'_1; \ldots; z'_N)$ satisfy $f(x) = f(x')$. Consider $y$ with $y_j = (1, \ldots, q-1, j)$ for each $j \in \{q, \ldots, N\}$. Then,

$$q\mathrm{SA}(x)_j = \frac{1}{q}\xi_j \text{ and } q\mathrm{SA}(x')_j = -\frac{1}{q}\xi_j.$$

Hence, $\|q\mathrm{SA}(x)_{j^*} - q\mathrm{SA}(x')_{j^*}\|_2 \ge \frac{2}{q}$. Because $f(x) = f(x')$,

$$\max \left( \|f(x) - q\mathrm{SA}(x)_{j^*}\|_2, \|f(x') - q\mathrm{SA}(x')_{j^*}\|_2 \right) \ge \frac{1}{q},$$