# OpenReview forum: "Representational Strengths and Limitations of Transformers"
_NeurIPS.cc/2023/Conference — NeurIPS 2023 poster_

### Official Review · Reviewer_CgC3 · 2023-07-06

**Soundness:** 3 good
**Presentation:** 3 good
**Contribution:** 3 good
**Rating:** 6
**Confidence:** 2

**Summary:**

This paper investigates the representation power of attention layers in transformer networks and compares them with other neural network architectures. The authors establish both positive and negative results on the benefits and limitations of attention layers, focusing on intrinsic complexity parameters such as width, depth, and embedding dimension. The positive results include the demonstration of a sparse averaging task where transformers scale logarithmically in the input size, compared to polynomial scaling in recurrent networks and feedforward networks. They also show the necessity and role of a large embedding dimension in transformers. On the negative side, they present a triple detection task where attention layers have linear complexity in the input size. However, they also provide variants of the task that can be efficiently solved by attention layers. The paper's contributions include the formalization of computational limits using communication complexity, the establishment of the representational capabilities and limitations of self-attention units, and the demonstration of the impossibility of computing Match3 using standard multi-headed attention layers in an efficient manner.

**Strengths:**


- This paper brings some original contributions to the understanding of attention layers in transformer networks. It provides a mathematical analysis of the representation power of attention layers and compares them to other neural network architectures. The paper introduces some interesting tasks, such as sparse averaging, pair matching, and triple matching, to evaluate the capabilities and limitations of attention layers. This approach introduces fresh perspectives on the benefits and deficiencies of attention layers and offers valuable insights into their role in deep learning.

- The paper demonstrates a high level of quality in terms of its analysis and proofs. It presents rigorous mathematical formulations and provides formal definitions for the tasks and architectures under study. The proposed theorems and conjectures are well-supported and backed by detailed proofs, showcasing the authors' expertise in the subject matter. The use of communication complexity techniques adds depth and reliability to the analysis. The paper also includes supplementary proofs and explanations in the appendices, further enhancing the quality and comprehensiveness of the work.

- The paper addresses an important gap in the literature by providing a mathematical analysis of the benefits and limitations of attention layers in transformers. The findings have implications for the design and optimization of deep learning models, especially in natural language processing and other sequential tasks. The identification of tasks that highlight the strengths and weaknesses of attention layers can guide future research in developing more efficient and effective architectures. The paper also raises intriguing conjectures, such as the impossibility of efficiently computing Match3, which can inspire further investigations and spark valuable discussions in the research community.

**Weaknesses:**


- While the paper provides rigorous mathematical analysis and proofs, it lacks empirical evaluation of the proposed tasks and architectures. Including experiments using real-world datasets could provide practical validation of the theoretical findings and further strengthen the paper's conclusions. Empirical evaluation could also provide insights into the computational efficiency and generalization performance of attention layers compared to other architectures.

- Providing concrete examples of how the theoretical findings can be applied to practical deep learning problems, such as natural language processing or computer vision tasks, would enhance the paper's relevance and impact.

- While the paper is generally well-written, the mathematical formulations and proofs can be complex and challenging to follow for readers without a strong background in the subject area. Simplifying and clarifying the presentation of the mathematical concepts and providing more intuitive explanations or examples could improve the accessibility of the paper for a wider audience.

**Questions:**

Please see above comments.

**Limitations:**

Not applicable.

---

> ### Author Rebuttal · Authors · 2023-08-09
>
> > While the paper provides rigorous mathematical analysis and proofs, it lacks empirical evaluation of the proposed tasks and architectures. Including experiments using real-world datasets could provide practical validation of the theoretical findings and further strengthen the paper's conclusions. Empirical evaluation could also provide insights into the computational efficiency and generalization performance of attention layers compared to other architectures.
>
> Empirical evaluations on real-world datasets cannot establish fundamental limits or asymptotic separations between different Transformer architectures. While we agree that that computational and statistical properties of transformers can be rigorously studied with empirical methods, these are not the focus of this work, which focuses on approximation capabilities and fundamental limitations.
>
> We do include brief experiments in Appendix D, which, while not comprehensive, suggest that transformers have a favorable inductive bias for learning qSA from randomly drawn samples, in contrast to other standard neural architectures like MLPs and LSTMs, which both overfit the training dataset with disastrous generalization. We chose not to emphasize these experiments due to our theoretical focus and the space limitations, but the extra space allocated to the camera-ready version would allow us to present this in the main body.
>
> > Providing concrete examples of how the theoretical findings can be applied to practical deep learning problems, such as natural language processing or computer vision tasks, would enhance the paper's relevance and impact.
>
> The theory of transformers is not yet at this stage; we are still laying the foundations. (This is similar to early theoretical work on MLPs from Kolmogorov and Arnold from 1957, which characterized a broad class of multivariate functions as a superposition of continuous univariate functions and forms a foundation for future work on the universal approximation of 2-layer MLPs. Recent generalization work about MLPs with more practical implications follows several years of approximation-theoretic work on foundational issues.)
>
> > While the paper is generally well-written, the mathematical formulations and proofs can be complex and challenging to follow for readers without a strong background in the subject area. Simplifying and clarifying the presentation of the mathematical concepts and providing more intuitive explanations or examples could improve the accessibility of the paper for a wider audience.
>
> We are happy to address clarity issues in the presentation. Can you give any specific pointers?

---

### Official Review · Reviewer_SrVL · 2023-07-06

**Soundness:** 4 excellent
**Presentation:** 3 good
**Contribution:** 2 fair
**Rating:** 4
**Confidence:** 4

**Summary:**

The paper presents the representational strength and some limitations of the transformer architecture.
1. Strength: separation between a unit of self-attention and a one-hidden layer neural or a recurrent neural network.
The authors present a task where the complexity of the latter networks scale with N (number of tokens) and the self-attention does not.
2. Limitation: the task Match2 can be computed with a self-attention unit that scales with the input dimension d, yet, a modification of the task, Match3, cannot be computed with a single transformer layer. Match3 can be computed with a standard and modified transformer model. The first makes assumptions over the input the second model modifies the self-attention module.



**Strengths:**

The paper is very well written. I could easily understand and follow all the definitions and the Theorems presented in the paper's main text.

**Weaknesses:**

The weakness of this paper lies in its relevance to the understanding of transformer networks.

The paper presents problems qSA, Match2, and Match3, whose relevance to understanding neural networks in practice is unclear. This is even stated in the paper: "Future work by linguists, theoretical computer scientists, and empirical NLP practitioners could assess how foundational our primitives are and study whether there are any practical triple-wise problems that transformer models fail to solve."

I think that when studying theoretical aspects of neural networks, it is the responsibility of the authors to motivate their theoretical results and why their assumptions or framework are relevant. Otherwise, such theoretical content belongs in a mathematical venue.

Minor comment:
When reading Section 1.1 for the first time, I could not follow the authors' intentions/message. Only after carefully reading the rest of the text and definitions could I understand it. As written now, I think it does not convey information properly for someone just interested in having a quick general idea of the results.


**Questions:**

I would be happy to change my mind about the paper since the paper is written technically very well. Better than most theoretical papers I encounter!

Can the authors at least give some intuitive explanations why they think the tasks presented in the paper are relevant to NLP or any other ML task? Maybe a small experimental study on a small transformer network and show that the functions qSA, Match2, and Match3 somewhat resemble the functions the transformer networks actually compute?

More so, through the years, I haven't encountered a single theory paper about the representational power of neural networks that actually sheds any light on understanding neural networks better in the practical sense of explaining their performance (other than the original universal approximation paper). For example, one can study the VC dimension of different architectures, but this does not bring us any step closer to understanding neural networks in practice.

In its current form, I think the paper is suited to a pure theoretical venue such as a math journal or a theoretical CS conference. To make it more suitable for an ML venue, a reasonable amount of effort is required to motivate theoretical setup and questions. In that case, it's likely that the scope of the paper would not fit a conference paper but in a journal.


**Limitations:**

The authors adequately addressed the limitations of their work.

---

> ### Author Rebuttal · Authors · 2023-08-09
>
> > Minor comment: When reading Section 1.1 for the first time, I could not follow the authors' intentions/message. Only after carefully reading the rest of the text and definitions could I understand it. As written now, I think it does not convey information properly for someone just interested in having a quick general idea of the results.
>
> We appreciate this comment, and upon revisiting the section, we recommend a few paragraphs the beginning of the section to clarify our contributions before introducing specific details and notation about tasks. Please see our response to reviewer iCyX for our planned revision of Section 1.1 in the camera-ready version.
>
> > Can the authors at least give some intuitive explanations why they think the tasks presented in the paper are relevant to NLP or any other ML task? Maybe a small experimental study on a small transformer network and show that the functions qSA, Match2, and Match3 somewhat resemble the functions the transformer networks actually compute?
>
> Our intuition was primarily shaped by reading papers that analyzed the self-attention matrices on NLP tasks, such as the following:
>
> Clark K, Khandelwal U, Levy O, Manning C. What Does BERT Look At? An Analysis of BERT's Attention. In ACL 2019.
>
> Rogers A, Kovaleva O, Rumshisky A. A Primer in BERTology: What We Know About How BERT Works. In ACL 2020.
>
> Chen N, Sun Q, Zhu R, Li X, Lu X, Gao M. CAT-probing: A Metric-based Approach to Interpret How Pre-trained Models for Programming Language Attend Code Structure. In EMNLP 2022.
>
> The qSA task was motivated by the fact that most self-attention matrices appear to be either wide-scale averages over a large fraction of tokens (where most inputs average over the same collection of other inputs and the identity of the token has little bearing on which elements it is associated with), or sparse matrices where different inputs map to different averages of outputs. These often correspond to intuitive linguistic relationships, such as those linking antecedents and coreferences. We were interested in the limitations of this sparse collection of individual linkages, and we crystallized sparse averaging as a way to make this problem concrete.
>
> As noted by Rogers et al, grammatical structures and syntax are directly encoded in self-attention matrices, which can in turn be decomposed into sentence diagrams or trees over multiple layers. We see Match2 as a fundamental unit of that phenomenon, since trees are easily decomposed into combinations of pairwise matches. We also see a more direct motivation for Match2 in co-reference resolution. On the other hand, we see Match3 as an operation that does appear in standard NLP tasks, and that when it does, it's analogous to Match3Assist or Match3Local. Rather, we considered Match3 to represent a family of sequential learning problems that we expect to be much more difficult to solve than real-world language problems.
>
> > I think that when studying theoretical aspects of neural networks, it is the responsibility of the authors to motivate their theoretical results and why their assumptions or framework are relevant. Otherwise, such theoretical content belongs in a mathematical venue.
>
> We agree with the general standard---applicable to all papers and not just "theoretical papers"---of providing motivation and justifying assumptions and frameworks, and our paper does meet this standard. The motivation to study transformers is given throughout the introduction of the paper, along with specific justification for our analysis framework and scaling regimes. Understanding fundamental capabilities and limitations of transformers is of interest to researchers studying these models, and the types of separation results we proved have a long history of being presented at machine learning venues, including NeurIPS, over the past few decades.
>
> We don't agree that "theoretical papers" are subject to a different standard for NeurIPS compared to other types of papers, as the [NeurIPS Call for Papers](https://neurips.cc/Conferences/2023/CallForPapers) explicitly welcomes learning theory papers without stipulating any such qualifications.
>
>
> > More so, through the years, I haven't encountered a single theory paper about the representational power of neural networks that actually sheds any light on understanding neural networks better in the practical sense of explaining their performance (other than the original universal approximation paper). For example, one can study the VC dimension of different architectures, but this does not bring us any step closer to understanding neural networks in practice.
>
> While it is true that theory has failed to provide an end-to-end story for deep learning (e.g., why gradient descent applied to transformers can lead to good question-answering abilities), it has provided many useful *suggestions* and *mental models* for reasoning.  For example, capacity theory (such as generalization bounds) motivate the use of regularization and weight decay, which are still used in modern architectures.  The classical theory of universal approximation, while lacking effective bounds and guidance for architecture selection, do sanity check the enormous representational power of deep networks.  In our case, while our theorems are purely representational, in appendix D we verify empirically that qSA is a problem efficiently learned empirically by transformers, but not by other architectures (as evidence by their poor *test* error); tying this back, while perhaps the reviewer feels our work does not have explicit empirical suggestions, it motivates a problem which does capture some empirical benefits of transformers over their predecessors.

---

> > ### Comment · Reviewer_SrVL · 2023-08-14
> >
> > Thank you for your response!
> >
> >
> >
> > **"The qSA task was motivated by the fact that most self-attention matrices appear to be either wide-scale averages over a large fraction of tokens (where most inputs average over the same collection of other inputs and the identity of the token has little bearing on which elements it is associated with)"**
> >
> > Can you refer me to the specific paper and the location where such a statement is made/supported?
> >
> > **"As noted by Rogers et al, grammatical structures and syntax are directly encoded in self-attention matrices, which can in turn be decomposed into sentence diagrams or trees over multiple layers. We see Match2 as a fundamental unit of that phenomenon, since trees are easily decomposed into combinations of pairwise matches."**
> >
> > I'm unfamiliar with sentence diagrams or trees over multiple layers (but I'm happy to learn). Can you help me and refer me to the exact location of Rogers et al. where your claim is supported (since it's a survey of over 150 studies)?
> >
> > **We agree with the general standard---applicable to all papers and not just "theoretical papers"---of providing motivation and justifying assumptions and frameworks, and our paper does meet this standard. The motivation to study transformers is given throughout the introduction of the paper, along with specific justification for our analysis framework and scaling regimes. Understanding fundamental capabilities and limitations of transformers is of interest to researchers studying these models, and the types of separation results we proved have a long history of being presented at machine learning venues, including NeurIPS, over the past few decades.**
> >
> >
> >  I totally agree that you mention the above in your text! I only had a problem with the lack of support for the tasks you considered.
> >
> > I did not see references for
> >
> > Clark K, Khandelwal U, Levy O, Manning C. What Does BERT Look At? An Analysis of BERT's Attention. In ACL 2019.
> >
> > Chen N, Sun Q, Zhu R, Li X, Lu X, Gao M. CAT-probing: A Metric-based Approach to Interpret How Pre-trained Models for Programming Language Attend Code Structure. In EMNLP 2022.
> >
> > These works provide some motivation for the tasks, as you replied in your answer. Did I miss it in the main text? or some other form of explanation?
> >
> > **We don't agree that "theoretical papers" are subject to a different standard for NeurIPS compared to other types of papers, as the NeurIPS Call for Papers explicitly welcomes learning theory papers without stipulating any such qualifications.**
> >
> > I agree with you about the standard for NeurIPS.  It was under the assumption that the task at hand was unmotivated and unrelated to practical transformers.
> >
> >
> >
> > **in appendix D we verify empirically that qSA is a problem efficiently learned empirically by transformers, but not by other architectures (as evidence by their poor test error); tying this back, while perhaps the reviewer feels our work does not have explicit empirical suggestions, it motivates a problem which does capture some empirical benefits of transformers over their predecessors.**
> >
> > After a quick glance, and I might be wrong, it seems the task you empirically examined is very artificial.
> > Might it be the case that with actual sentences embedded with word2vec, for example, a fully connected network (or other non-transformer) can learn qSA?
> >
> > If not, I think this would establish your case much better because right now, it might be that fully connected networks fail because they are presented with an unnatural and unrealistic distribution.

---

> > > ### Author Response · Authors · 2023-08-20
> > >
> > > Thank you for your thoughtful review and comments.
> > >
> > > **Regarding your query about self-attention matrix sparsity patterns:**
> > > The appearance of self-attention matrices in Figure 3 of Rogers et al and Figure 1 of Likhosherstov et al suggest that the outputs of softmax attention units resemble either sparse matrices with complicated patterns or low-rank non-sparse matrices. Additionally, the sparsity assumption made in Likhosherstov et al aligns with this observation.
> > >
> > > Likhosherstov V, Choromanski K, Weller A. On the Expressive Power of Self-Attention Matrices
> > >
> > > **On the encoding of grammatical structures in self-attention matrices by Rogers et al:**
> > > We direct your attention to Section 4.3 of Rogers et al. In particular, the work by Hewitt and Manning using structural probes has identified an iterative encoding of syntactic trees in embeddings of intermediate layers of ELMo and BERT models.
> > >
> > > Hewitt J, Manning C. A Structural Probe for Finding Syntax in Word Representation.
> > >
> > > **About the references Clark et al (2019) and Chen et al (2022):**
> > > We regret that these papers were not cited in our submission, since we opted to present the work with a primary focus on the theoretical results. While indirectly, these papers provided inspiration for the tasks we chose to construct, and we think that the inclusion of these citations when introducing the tasks and sharing open problems will improve readability. The initial inclusion or exclusion of certain references was to maintain clarity and focus, but we acknowledge that broadening our citation scope might enhance our paper's context and readability. We will take this into account in our revisions.
> > >
> > > **Regarding the qSA task:**
> > > The purpose of the qSA task is to crystallize the intrinsic capabilities and limitations of transformers, and we acknowledge that from the standpoint of NLP research, the task is artificial. The empirical results we presented aimed to demonstrate that transformer architectures are particularly suited for this task, as previously requested by the reviewer; the intention of these experiments is not to establish a relationship to linguistic tasks.
> > >
> > > While we appreciate your point about the potential capabilities of other architectures under different embeddings, our focus was to elucidate the unique strengths of transformers by providing a simple and concrete task that differentiates the representational abilities of different architectures. It's unclear to us why a word2vec embedding would help a fully connected network learn qSA (when the qSA task is not actual sentences) or what the experiment would reveal about the limitations of different architectures.
> > >
> > > Thank you for your feedback, and we hope this addresses your concerns.

---

### Official Review · Reviewer_jpuU · 2023-07-07

**Soundness:** 3 good
**Presentation:** 2 fair
**Contribution:** 3 good
**Rating:** 6
**Confidence:** 2

**Summary:**

This paper focuses on the representational capabilities of attention layers in transformer models and showcases both the strengths and limitations. On one hand, the paper proves that attention layers excel at the presented sparse averaging task compared to RNNs and FNNs. On the negative side, they have complexity scaling linearly in the input size for the triple detection task. The paper theoretically showcases the strengths and limitations of the expressivity of attention layers, especially the role of embedding dimension of the attention layer.

**Strengths:**

To my knowledge, the presented theoretical results are novel in the study of transformers and their representational capabilities. However, this work is out of my expertise and I cannot go into depth about the significance of the work. I do appreciate the authors in providing some empirical evidence about the theoretical findings of the work.

**Weaknesses:**

As said previously, this is not within my area of expertise. I am however interested in how relevant/connected are the proposed tasks (e.g. sparse averaging and triple detection) to empirical studies such as language or visual data modeling. I think the work would be improved if more insights could be provided as to how the theoretical results can be connected to real-world practices.

**Questions:**

My questions and suggestions are listed above.

**Limitations:**

This is a theory paper and the authors provided open directions for future work.

---

> ### Author Rebuttal · Authors · 2023-08-09
>
> > As said previously, this is not within my area of expertise. I am however interested in how relevant/connected are the proposed tasks (e.g. sparse averaging and triple detection) to empirical studies such as language or visual data modeling. I think the work would be improved if more insights could be provided as to how the theoretical results can be connected to real-world practices.
>
> We agree that it would be very interesting to connect the tasks to real-world problems and practices. However, it is also independently interesting to establish fundamental capabilities and limitations of transformers on mathematically precise settings. See the response to reviewer SrVL for an explanation of our intuition for selecting these tasks.
>
> In short, the sparse averaging task was inspired by analyses of the sparsity patterns of attention matrix softmax outputs. We study Match2 because we see a connection between Match2 is a useful primitive for language tasks like coreference resolution and because tree-like grammars can be reconstructed by applying Match2 multiple times; in contrast, Match3 is an analogous primitive that transformers apparently fail to represent without contextual clue (e.g. Match3Assist or Match3Local).  Standard literature on coreferences may be found [at this page hosted by the stanford NLP group](https://nlp.stanford.edu/projects/coref.shtml); we will revise our work with this discussion and appropriate citations.

---

### Official Review · Reviewer_iCyX · 2023-07-07

**Soundness:** 2 fair
**Presentation:** 2 fair
**Contribution:** 3 good
**Rating:** 5
**Confidence:** 3

**Summary:**

This paper mainly investigates the inductive biases of attention-based models. They propose three computational tasks that show the limitations of Transformers, namely, sparse averaging, pair matching, and triples-matching. Specifically, they analyze the representational power of embedding dimensions and show that the sparse averaging task scales more favorably in the input size as opposed to the other two tasks. Various proofs are provided to support the investigations and conclusions.

**Strengths:**

* Interesting and relevant topic on interpreting the Transformer computations and learning progress.
* Good connections with relevant prior works in setting the backgrounds for inspiring the tasks and their practical implications proposed

**Weaknesses:**

* Initially the notation is a bit confusing, especially in section 1.1 that details the contributions. Variables $y$ and $z$ should be more clearly defined and explained when mentioning results of the theoretical analysis.
* The intro can maybe be reworked to slowly work in the details and notation of the method instead of initially presenting the mechanisms that the different tasks enable

**Questions:**

* It seems somewhat obvious that the representational capacity of performant Transformeres is related to the embedding space and operations for comparison performed on thus. Have the authors experimented with different contexts aside from the embedding space?

**Limitations:**

Yes.

---

> ### Author Rebuttal · Authors · 2023-08-09
>
> > Initially the notation is a bit confusing, especially in section 1.1 that details the contributions. Variables and should be more clearly defined and explained when mentioning results of the theoretical analysis.
>
> Upon a closer reading, we see that we use a large amount of condensed English and notation, which hinders a first reading. We will gladly refine and expand this section, using some of the extra space granted with the extra camera ready page. Would the reviewer like to point out any specific frustrations in Section 1.1?
>
> More broadly, we think that adding a few paragraphs the beginning of the section to clarify the paper's contributions before introducing the task formulations would help remedy the issues identified by this reviewer and SrVL.
>
> 1. Brief description of the transformer architecture
>     - Simple mathematical formulation of a self-attention unit (without MLPs to keep it simple)
>     - Overview of how transformers are assembled by composing these units in parallel and in series
>     - Introduction of all relevant architectural resources and their corresponding variables: number of heads, depth, embedding dimension, bit-precision
> 2. Overview of our main findings (without getting into details of the problem)
>     - Goal of the paper is to identify tasks that cleanly separate the abilities of different neural architectures with a focus on transformers: what self-attention can do that other models cannot, how embedding dimension modulates approximation power, fundamental limitations of self-attention can do
>     - The methodology of the paper is to introduce such tasks and prove upper and lower bounds the resources necessary for neural architectures to solve those tasks
>
> > The intro can maybe be reworked to slowly work in the details and notation of the method instead of initially presenting the mechanisms that the different tasks enable
>
> > It seems somewhat obvious that the representational capacity of performant Transformeres is related to the embedding space and operations for comparison performed on thus. Have the authors experimented with different contexts aside from the embedding space?
>
> We agree that it is intuitive that results such as ours should be true, but they have not been proved for transformers before our work. Our investigation also reveals fundamental benchmark problems (e.g., qSA, Match2) that we expect to be useful in future studies of transformers and related architectures.
>
> We abstracted away many aspects of transformers and embeddings, and this certainly poses interesting questions for future work. (Our particular choices are in part motivated by the practical scaling trends discussed at the top of Page 2 of our paper.) We would very much appreciate suggestions for other aspects of embeddings that might be amenable to mathematical analysis.

---

### Official Review · Reviewer_bmhz · 2023-07-28

**Soundness:** 4 excellent
**Presentation:** 3 good
**Contribution:** 4 excellent
**Rating:** 7
**Confidence:** 4

**Summary:**

The authors study the representational power of transformers. They quantify how the transformers are superior to other neural network architectures, as well as the limitation of the transformer architecture. They focus on the $q$-sparse averaging (qSA) task which amounts to  averaging $d$-dimensional input vectors over subsets of size $q$ over a set of size $N$. They show by an explicit construction a unit of self-attention with $m \geq q$ dimensional embedding can approximate qSA. Using tools from communication complexity between two parties, they show fully connected networks require $\Omega(Nd)$ hidden layers, and recurrent neural networks require $\Omega(N)$ bits of information. They further show that standard transformers can approximate functions that intrinsically depends on input pairs, while failing to approximate functions that depends on triplets instead. Finally, they propose a variant of transformers that can approximate the triplet-functions.

**Strengths:**

- The authors leverage the communication complexity to show that the self attention layers add more representational capacity as compared to fully connected and recurrent neural networks.

- They use Match2 and Match3 problems to show rigorously that 2nd-order functions represent the threshold of efficient approximability for transformers.

- The variants of Match3 problems hint at transformers leveraging some local structures present in the data to go beyond pair matching. This reconciles the Match3 impossibility result with the much presumably higher order successes of transformer models.

- They additionally show higher order transformers can solve the higher order function approximations.





**Weaknesses:**

- The fixed precision (Theorem 2) result still uses precision that needs to grow with the data size and approximation parameters. Although it is a good first step, it leaves open the approximability with constant precision (which is used in most practical cases).

- The ``third-order tensor self-attention" is an important claim in the paper, but it is fully omitted in the main paper. It will be good to at least introduce the basic ideas in the main paper.

**Questions:**

- Given a fixed constant precision $p$ what is the bottleneck in creating the transformers that can approximate q-SA? What if $z_i$s are themselves restricted to some $O(p)$ precision?

- Is the use of communication complexity in the approximability of Transformers (and broader Neural networks) novel in this work?

**Limitations:**

It is hard to foresee any potential negative societal impact of this theoretical work.

---

> ### Author Rebuttal · Authors · 2023-08-09
>
> > The fixed precision (Theorem 2) result still uses precision that needs to grow with the data size and approximation parameters. Although it is a good first step, it leaves open the approximability with constant precision (which is used in most practical cases).
>
> We agree that constant precision is an interesting case to consider, and we thank the reviewer for raising this as a potential area of improvement.
>
> When reassessing our positive result in Theorem 3, we realized that the bit precision analysis can be improved to $p = \Omega(\log(q\log(N) \epsilon))$, which should partially address the reviewer's question by reducing the dependence on $N$ to doubly-logarithmic. In brief, the sharpening proceeds by augmenting the proof with the following analysis:
> - Under $p = \Theta(\log (q\log(N) / \epsilon))$-bit precision, each query vector $w_y$ can be quantized with a $p$-bit floating point vector that that approximates it to accuracy $\text{poly}(\epsilon / (q \alpha))$.
> - Hence, we can ensure that the computed inner products satisfy $\langle u_{i'}, w_y \rangle \in [1 - \text{poly}(\epsilon / (q \alpha)), 1+ \text{poly}(\epsilon / (q \alpha))]$ if $i' \in y$ and $\langle u_{i'}, w_y \rangle \in [\frac12 - \text{poly}(\epsilon / (q \alpha)), \frac12 + \text{poly}(\epsilon / (q \alpha))]$.
> - Propagating this change forward, we can ensure that $\text{softmax}(\phi(X) QK^T \phi(X)^T)\_{i, i'} \in [\frac{(1 - \epsilon/2)}q, \frac{(1 + \epsilon/2)}q]$ if $i' \in y_i$ and $\text{softmax}(\phi(X) QK^T \phi(X)^T)\_{i, i'} \leq \frac{\epsilon}{2N}$ otherwise. While $\log N$ bits are needed to accurately represent the latter quantity, a lower bit-complexity would simply round the term to zero, which does not hinder the ability of $f(X)$ to approximate $\text{qSA}(X)$.
>
> The same quality of approximation is thus recovered without requiring $O(\log N)$-bit precision.
>
> However, we would also like to note that we think it's reasonable to consider a $p = \Theta(\log N)$ scaling for bit-precision. Trained self-attention matrices frequently compute non-sparse averages over all inputs (see Figure 13 in the supplement of Jumper et al, 2021), where the softmax outputs approximately $\frac1N$ for each input token. Since transformer implementations use floating-point arithmetic, one can assume that the precision of the floating point is be large enough to ensure that $\frac1N$ is not rounded to zero.
> On a similar note, we do not expect our bounds to work with less than $O(\log q)$-bit precision, since the aim is to approximate a function that computes an average over $q$ elements.
>
> Jumper J, Evans R, Pritzel A, Green T, Figurnov M, Ronneberger O, Tunyasuvunakool K, Bates R, Žídek A, Potapenko A, Bridgland A. Highly accurate protein structure prediction with AlphaFold. Nature. 2021.
>
>
> > Given a fixed constant precision what is the bottleneck in creating the transformers that can approximate q-SA? What if z_is are themselves restricted to some precision?
>
> See above.
>
> > The "third-order tensor self-attention" is an important claim in the paper, but it is fully omitted in the main paper. It will be good to at least introduce the basic ideas in the main paper.
>
> Thank you for the suggestion; we'll do just that. (The additional content page in the camera-ready should more than suffice.)
>
> > Is the use of communication complexity in the approximability of Transformers (and broader Neural networks) novel in this work?
>
> We have not seen communication complexity used in the context of Transformers before, but it has been used to prove lower bounds for neural networks (and circuits/formulas) before. We'll make sure to cite these and other works in the camera-ready version.
>
> Karchmer M, Wigderson A. Monotone circuits for connectivity require super-logarithmic depth. In Proceedings of the Twentieth Annual ACM Symposium on Theory of Computing, 1988.
>
> Martens J, Chattopadhya A, Pitassi T, Zemel R. On the representational efficiency of restricted Boltzmann machines. Advances in Neural Information Processing Systems, 2013.
>
> Vardi G, Reichman D, Pitassi T, Shamir O. Size and depth separation in approximating benign functions with neural networks. In Conference on Learning Theory, 2021.

---

> > ### Comment · Reviewer_bmhz · 2023-08-12
> > **Response to Rebuttal**
> >
> > I thank the authors for their helpful comments. The lowering of the precision required for their bounds is welcome. The argument sounds reasonable (although not fully verified). Overall my concerns are addressed. These concerns did not influence my evaluation much. I will maintain my score.

---

### Author Rebuttal · Authors · 2023-08-09

We thank the reviewers for their detailed and thoughtful feedback on our submission. We are grateful that the reviewers largely appreciated the strength and value of our fundamental theoretical contributions while identifying areas of improvement. We agree that some aspects of the presentation of the paper can be improved, and our replies to specific comments detail our plan to do so. If any questions are unanswered or our responses are unclear, we would appreciate the chance to engage further with our reviewers.

Briefly, the key points of our response are the following:

1. **Precision analysis:** Reviewer bhmz asked about whether our results can be adapted to constant bit precision. We appreciate the constructive feedback, and while considering it, realized that our analysis of the bit precision of our positive sparse-averaging result can be significantly sharpened (from logarithmic to doubly logarithmic dependence on sequence length). We explain in detail in our response below why we believe this bit precision bound to be nearly optimal.

2. **Clarity and notational issues:** Reviewers iCyX and SrVL both noted a difficulty in understanding the framing of our contributions in the introductory Section 1.1, in particular due to the mathematical notation. We appreciate the reviewers calling attention to this issue, and we will add a few paragraphs (outlined in the response to iCyX) to clarify our contributions without dense notation.

3. **Omission of third-order tensor self-attention:** We acknowledge the oversight that third-order tensor attention was insufficiently discussed in the main paper body raised by reviewer bhmz. We regret the omission, and we intend to remedy it with the additional page permitted for the camera-ready version.

4. **Relevance to practical tasks:** Reviewers jpuU, SrVL, and CgC3 requested clarification on our motivation for choosing the qSA, Match2, and Match3 problems and the relevance of our work to empirical training of transformers. While our goal was to define tasks that clearly delineate the capabilities of different architectures, we agree that additional context would help readers understand our contributions. We outline our intuitions about why these tasks are relevant in the response to SrVL, and we call attention to our brief empirical results in Appendix D.

Once again, we are grateful for the time and effort put into reviewing this submission, and we firmly believe that these comments will strengthen the clarity and motivation of our manuscript.

---

### Decision · Program_Chairs · 2023-09-21

**Decision:**

Accept (poster)

**Comment:**

This paper aims to study the representational power of the attention layer -- a key component of the transformer architecture. The authors present a task, namely q-sparse averaging, which a single-layer transformer can approximate with the embedding dimension scaling logarithmically in the sequence length $N$. In contrast, fully connected networks or RNNs require network size to scale polynomially in $N$. The authors also discuss a pair detection task Match2 which the attention layer can efficiently approximate. The authors then highlight the limitation of the attention layer by considering a triple detection task Match3 which requires the attention layer to scale its parameters (number of heads or embedding dimension) polynomially in N. Interestingly, the author also identified structured versions of Match3 which attention layer can efficiently approximate.

Overall, the reviewers agree that this paper makes novel contributions that are technically sound. The paper leverages tools from communication complexity to study the fundamental limits of transformers which will inspire other follow-up works.

One of the key concerns that multiple reviewers raised during the review process was that it was not clear how the tasks considered in this paper are connected with the practice. In their response, the authors pointed out many empirical studies in NLP that served as a motivation for the task considered in this work. Although there is no one-to-one connection with the tasks encountered in practice, the tasks studied in this submission do capture some aspects of the real-life tasks. This paper makes a good first step towards rigorously understanding the representational limits of transformers which many researchers will find interesting. I am sure that this paper will lead to follow-up works that will explore a rigorous analysis of the tasks that are more closely aligned to those encountered in practice.

One of the reviewers had asked to improve the bit precision requirement for Theorem 2. The authors did provide a sketch to improve their analysis and improve the precision requirement from logarithmic to doubly logarithmic.

The reviewers made multiple suggestions to improve the presentation of the paper which the authors have promised to incorporate in the revised version. In particular,

1) Please improve the presentation in Section 1.1.
2) Please include citations to the references that came up during the discussion with the Reviewer SrVL and provide a discussion on how different tasks considered in this work are (partially) inspired by either the observations in practical Transformer models or sub-tasks encountered in various real-life NLP settings.
3) Please highlight both higher-order attention and your experiments on the synthetic qSA task in the main body of the paper.